# Mask-inspired moisture-transmitting and durable thermochromic perovskite smart windows

Sai Liu [1,9], Yang Li[2,3,4,9], Ying Wang [5], Yuwei Du[1], Kin Man Yu[5], Hin-Lap Yip [1,6,7], Alex K. Y. Jen [6,7,8], Baoling Huang [4] ✉ & Chi Yan Tso [1] ✉

Thermochromic perovskite smart windows (TPWs) are a cutting-edge energy-efficient window technology. However, like most perovskite-based devices, humidity-related degradation limits their widespread application. Herein, inspired by the structure of medical masks, a unique triple-layer thermochromic perovskite window (MTPW) that enable sufficient water vapor transmission to trigger the thermochromism but effectively repel detrimental water and moisture to extend its lifespan is developed. The MTPW demonstrates superhydrophobicity and maintains a solar modulation ability above 20% during a 45-day aging test, with a decay rate 37 times lower than that of a pristine TPW. It can also immobilize lead ions and significantly reduce lead leakage by 66 times. Furthermore, a significant haze reduction from 90% to 30% is achieved, overcoming the blurriness problem of TPWs. Benefiting from the improved optical performance, extended lifespan, suppressed lead leakage, and facile fabrication, the MTPW pushes forward the wide applications of smart windows in green buildings.

As a result of rapid urbanization, modern buildings are responsible for over 40% of the global primary energy consumption, causing over 30% of greenhouse gas emissions in cities[1,2]. With strict esthetic requirements for buildings, an extremely high window-to-wall ratio has become a characteristic of modern architecture[3]. However, both the high transmittance under intense sunlight and the high heat transfer coefficient of glass make windows the major source of heat loss/gain among all building envelopes[4]. Therefore, energy-saving smart windows whose solar transmittance can be dynamically regulated have recently attracted increasing attention to balance the goal of less energy consumption with the esthetic demand for more glazing. The

most widely studied smart windows are electrochromic and thermochromic windows[5–7]. Unlike electrochromic windows with active control, thermochromic windows can switch between bleached and colored states in response to the ambient temperature in a passive way without electrical input, further mitigating energy usage[8,9].

Recently, the thermochromism of metal halide perovskites was observed. The inherently low formation energy of thermochromic perovskites (T-Perovskites) enables the rapid transformation from a light-absorbing phase (hot state) to a transparent phase (cold state) as the temperature changes, which results in a color switch[10–13]. Due to the facile solution-based synthesis method, high optical contrast in the

[1]School of Energy and Environment, City University of Hong Kong, Tat Chee Avenue Kowloon Tong, Hong Kong, China. [2]State Key Laboratory of Fluid Power and Mechatronic Systems, School of Mechanical Engineering, Zhejiang University, Hangzhou, China. [3]Key Laboratory of Advanced Manufacturing Technology of Zhejiang Province, School of Mechanical Engineering, Zhejiang University, Hangzhou, China. [4]Department of Mechanical and Aerospace Engineering, The Hong Kong University of Science and Technology, Clear Water Bay, Kowloon, Hong Kong, China. [5]Department of Physics, City University of Hong Kong, Tat Chee Avenue, Kowloon Tong, Hong Kong, China. [6]Department of Materials Science and Engineering, City University of Hong Kong, Tat Chee Avenue Kowloon Tong, Hong Kong, China. [7]Hong Kong Institute for Clean Energy, City University of Hong Kong, Tat Chee Avenue Kowloon Tong, Hong Kong, China. [8]Department of Chemistry, City University of Hong Kong, Tat Chee Avenue Kowloon Tong, Hong Kong, China. [9]These authors contributed equally: Sai Liu, Yang Li. ✉e-mail: mebhuang@ust.hk; chiytso@cityu.edu.hk

visible region, and tunable optical and transition properties, T-Perovskites show high energy-saving potential and exceptional climate adaptability[14–20]. However, the poor environmental stability of T-Perovskites hinders their applicability. The degradation of T-Perovskite films is mainly triggered by defects located on the surface and at grain boundaries[10,21]. These defects are very sensitive to water and oxygen, especially when T-Perovskites are continuously subjected to high-humidity environments or water droplets[22]. At the same time, the toxic lead (Pb) in T-Perovskites will leak into the water, which could cause environmental pollution and threaten public health[23,24]. One common method to protect T-Perovskites is to seal them in a double-glazed window[12,15]. However, this method requires tight packaging, making the assembly of the window difficult, with a risk of leakage during long-term use. The other technical route is to reduce the dimensions of the T-Perovskite to 2D, as this would offer improved moisture stability due to the incorporation of organic spacers[25]. However, 2D T-Perovskites suffer from a high transition temperature ($T_c > 60\,°C$) and a long transition time ($t > 6\,h$)[26] and still cannot compete with their 3D counterparts, whose $T_c$ values are near room temperature and $t$ values are only several minutes[15,16,19]. In other words, the two conventional methods used to circumvent the stability issue of T-Perovskites bring about new problems. To date, there has been no report of a fundamental solution to the stability issue without sacrificing the optical and transition performance. Therefore, developing durable and water-repellent T-Perovskite windows with good optical and transition properties is imperative and urgent.

Currently, some of the most widely investigated T-Perovskites are hydrated T-Perovskites. Relying on the dissociation from and rebonding to the T-Perovskite layer of $H_2O$[27], the color can be reversibly switched between transparent $MA_4PbX_6\,2H_2O$ and colored $MAPbX_3$ upon heating/cooling (equation 1).

$$MAPbX_3 + 3MAX + 2H_2O \rightleftharpoons MA_4PbX_6 \cdot 2H_2O \qquad (1)$$

where X is the halide anion. Note that water vapor is the key to inducing the thermochromic effect, but excess water vapor or water droplets will degrade T-Perovskites. In addition, because excess methylammonium iodide (MAI) in the precursor influences the crystallization process, the resulting high surface roughness causes strong light scattering, which leads to an ultrahigh optical haze (>90%) and a blurry view through T-Perovskite windows[28]. Therefore, to simultaneously enable thermochromism of T-Perovskite films, extend their lifespan in a humid environment and reduce their optical haze, an alternative configuration for T-Perovskite smart windows is required.

Herein, inspired by the unique trilayer structure of medical masks, we propose a mask-inspired thermochromic perovskite smart window (MTPW), which consists of T-Perovskite on the bottom, a middle protection buffer layer and a top superhydrophobic layer, to achieve both high water vapor transmission and durability. This trilayer design significantly enhances the durability of the pristine TPW. The solar modulation ability ($\Delta\tau_{sol}$) of the MTPW remains above 20% during a 45-day aging test at ~60% relative humidity, with a reduction in the decay rate by 37 times relative to that of the pristine TPW. Furthermore, MTPW can immobilize lead ions, suppressing lead leakage by 66 times when the window is submerged in water. Interestingly, without any extra chemical treatment (e.g., antisolvent) for the T-Perovskite, compared to the pristine TPW, the MTPW exhibits a significant haze reduction from 90% to 30%, resulting in the visually clear T-Perovskite window. Thus, the trilayer coating improves rather than impairs the luminous transmittance ($\tau_{lum}$) in both the cold and hot states. Because of its facile solution-based fabrication process, greater moisture resistance and waterproof ability, the MTPW can also be scalable as a flexible window film, increasing its applicability. These advantages of the MTPW make it a promising and impactful material for green building technologies.

## Results
### Design and fabrication of the MTPW
The design of the MTPW aims to simultaneously tackle the four bottlenecks of TPWs: (i) poor durability against moisture and water; (ii) lack of water vapor breathability to enable color switching when covered with a protection layer; (iii) ultra-high optical haze and insufficient optical transparency caused by poor surface morphology; and (iv) toxic lead leakage problems when encountering water. During the COVID-19 pandemic, medical masks were widely used to prevent viral transmission[29], which inspired us to propose the MTPW design. A typical medical mask includes three layers, as shown in Fig. 1a. The outermost layer is composed of spunbonded fabric that repels water, preventing body fluid or blood spatter. The middle layer consists of a melt-blown fabric that serves as a filter to block most viruses. The innermost layer is composed of an absorbent material to absorb moisture from the wearer's breath[30]. As a result of filtration by these three layers, clean air can be safely breathed by humans. Similar to a medical mask, the MTPW also consists of three layers (Fig. 1b). The bottom layer is a T-Perovskite film deposited on a glass substrate. To protect the T-Perovskite and maintain a high window transparency, a transparent protection buffer layer is adopted to control the amount of water vapor on the T-Perovskite. Finally, the top layer is a superhydrophobic layer to effectively repel liquid water droplets. In this way, the T-Perovskite window appears to wear a mask that allows it to limit water vapor transmission in the right amount to trigger thermochromism, but block excess water vapor and water droplets, thus improving its durability. Photographs of the MTPW are shown in Fig. 1c. It is highly transparent in the cold state and becomes reddish brown at the hot state, maintaining the reversible thermochromic effect (Supplementary Movie 1).

Here, hydrated $MAPbI_{3−x}Cl_x$ (H-$MAPbI_{3−x}Cl_x$) is selected as the T-Perovskite due to its large $\Delta\tau_{sol}$ (>20%), low $T_c$ (<45\,°C) and short $t$ (<2\,min)[15]. When selecting the protection layer, three criteria must be considered. First, the layer must be highly transparent. Second, the material should be stable and resistant to harsh environments. We measured the refractive index ($n$) of the H-$MAPbI_{3−x}Cl_x$ perovskite. As shown in Fig. 1d, $n$ is ~2.0 in the visible wavelength range in the cold state and is even higher at the hot state, giving rise to a strong light reflection at the air-perovskite interface according to Fresnel's formula. This leads to the third criterion for the protection buffer layer that it should have an $n$ between that of the perovskite and the air to minimize the surface reflection. Based on these requirements, we selected a widely used inorganic oxide, silicon dioxide ($SiO_2$), with an $n$ of ~1.5 as the buffer layer. Moreover, to solve the optical haze problem, the buffer layer coating method needs to be considered. The high haze of a T-Perovskite is attributed to its rough surface, which influences the light propagation path, leading to diffuse transmission (Supplementary Fig. 1a). Therefore, an effective method to reduce haze is to fill the valleys with a buffer layer to smoothen the perovskite surface (details are analyzed in the next section). However, $SiO_2$ coatings are typically deposited by high-vacuum clean-room coating methods, such as chemical vapor deposition and physical vapor deposition. These methods require expensive deposition equipment and can only deposit a conformal layer on the rough surface, which is not helpful in smoothing the surface (Supplementary Fig. 1b₁ and Supplementary Fig. 2)[31,32]. Based on the above analysis, solution-processed perhydropolysilazane (PHPS) was adopted to prepare the silicon oxide layer for the MTPW. PHPS is a kind of silicone that consists of silicon and nitrogen atoms in its backbone ($SiH_2\text{-}NH$)[33]. Upon annealing in air, PHPS solidifies to yield a homogeneous inorganic $SiO_x/SiON_x$ film, demonstrating better barrier properties and lower sensitivity than polymers in humid environments. The transmittance of the PHPS-coated glass was found to be ~90%, which is almost as high as that of bare glass (Supplementary Fig. 3), and the complex refractive index of the fabricated PHPS in this study is shown in Fig. 1d. PHPS

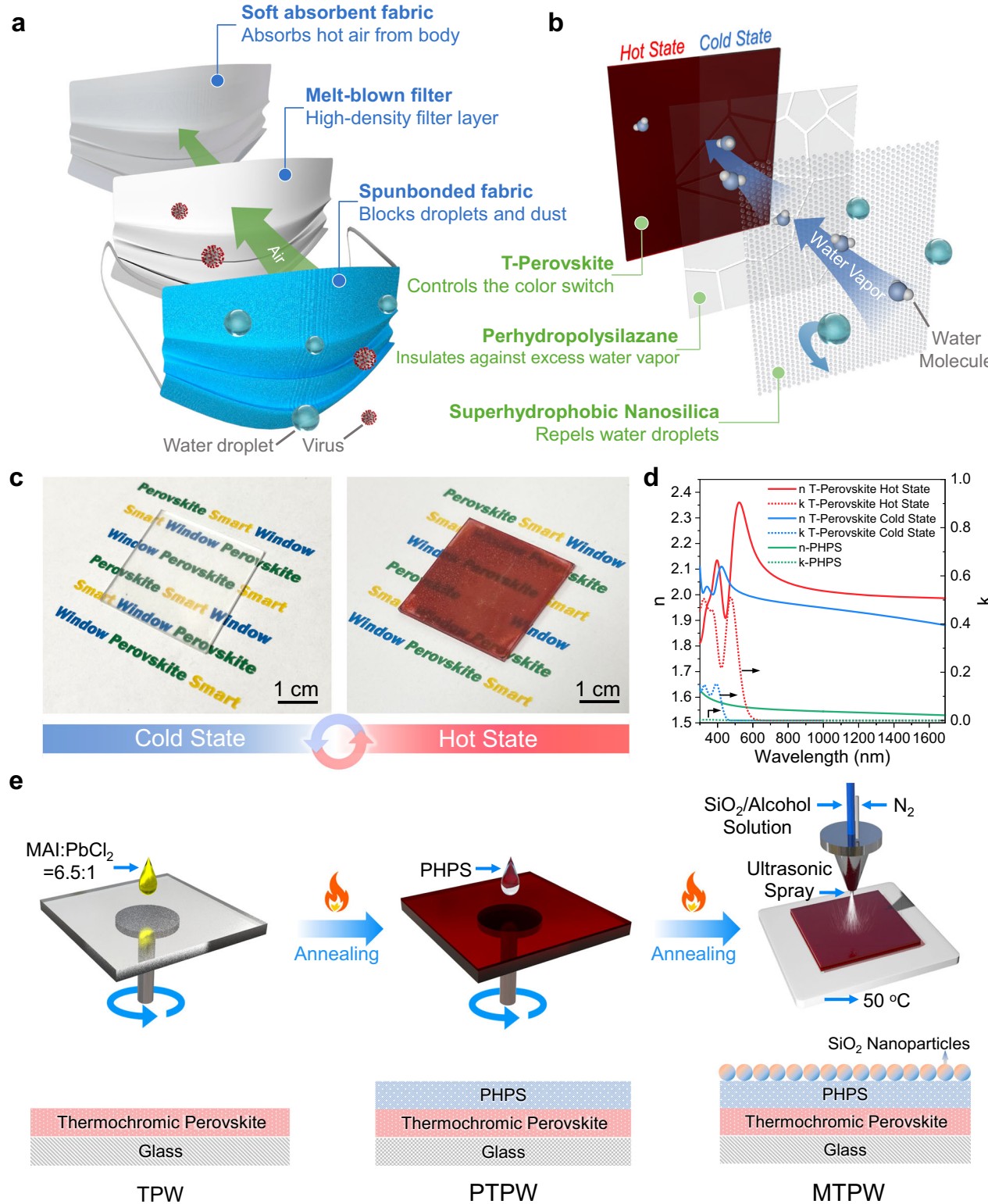

**Fig. 1 | Working principle of the mask-inspired thermochromic perovskite window (MTPW) and its fabrication process. a** Schematic of the trilayer structure and working principle of an antivirus medical mask. **b** Schematic of the trilayer structure and working principle of the MTPW for repelling water and excess water vapor. **c** Photographs of the MTPW in the cold and hot states. **d** Complex refractive index of T-Perovskite in the cold and hot states as well as the fabricated PHPS in this study. **e** Solution-based fabrication process of the MTPW. Firstly, a mixture of

methylammonium iodide (MAI) and lead chloride (PbCl₂) in a molar ratio of 6.5:1 is prepared. This mixture is then spin-coated onto a glass substrate, followed by an annealing process. Next, perhydropolysilazane (PHPS) is spin-coated on top of the thermochromic perovskite layer and subsequently annealed. Finally, a layer of silicon dioxide (SiO₂) nanoparticles is spray-coated onto the PHPS layer. Source data are provided as a Source Data file.

solution can be easily deposited using solution-based methods, such as spin-coating and blade-coating, thus smoothening the roughness of the T-Perovskite surface (Supplementary Fig. 1b$_2$). Note that previous studies on perovskite-based devices heavily relied on the antisolvent crystallization method to achieve high-quality films[34]. However, this method has yet to be successfully applied to T-Perovskites. Additionally, common antisolvents are toxic, and dripping antisolvents requires much operating experience[35,36]. Compared with the antisolvent method, in this study, without toxic chemical treatment or cumbersome operating techniques, a smooth T-Perovskite window was easily achieved by depositing a layer of PHPS, which is more environmentally friendly and convenient. Owing to these advantages, PHPS not only protects the T-Perovskite from excess water vapor but also improves its optical transmittance. Note that although PHPS can insulate against excess water vapor, it is not superhydrophobic, with a water contact angle of only ~99° (Supplementary Fig. 4)[37]. When the T-Perovskite window is exposed to bulk water or when water droplets accumulate on the surface, continuous water penetration can also damage the T-Perovskite. Therefore, similar to the hydrophobic spunbonded fabric in a mask that blocks body fluids, a superhydrophobic fluorinated nanosilica layer is applied on top of the MTPW to endow it with excellent water repellency.

As shown in Fig. 1e, an all-solution-based coating method was employed to fabricate the MTPW. The H-MAPbI$_{3-x}$Cl$_x$ precursor was prepared by mixing MAI and lead chloride (PbCl$_2$) in a molar ratio of 6.5:1. Then, the precursor was spin-coated on the glass, followed by annealing at 100 °C for 1 h in a glove box to form a 1.6 μm-thick T-Perovskite film (Supplementary Fig. 5a). To deposit the protective layer, PHPS was first dissolved in dibutyl ether to prepare a dilution. Dibutyl ether is a nontoxic and nonpolar solvent with a low dielectric constant of 3.1[38,39] and thus would not dissolve the bottom T-Perovskite. The PHPS dilution was spin-coated on the T-Perovskite and solidified by curing at 100 °C in air to form a ~1.8μm-thick film (Supplementary Fig. 5b). The moisture and oxygen in the air caused cleavage of the Si−N bond in the PHPS and the PHPS was gradually converted into homogeneous SiO$_x$/SiON$_x$ (Supplementary Fig. 6)[40]. As shown in Supplementary Fig. 7, as the curing time increased from 0 to 3 h, the Fourier transform infrared (FTIR) peaks corresponding to the N-H (3400 cm$^{-1}$), Si-H (2150 cm$^{-1}$) and Si-N (830 cm$^{-1}$) stretching vibrations decreased, while the Si-O peak near 1050 cm$^{-1}$ increased. The thickness of the PHPS layer influences the protection ability, optical performance, and transition time of the smart window. Note that while a thinner layer of PHPS results in a faster color switch process (Supplementary Fig. 8), the poor surface coverage on the T-Perovskite leads to weaker protection ability and poor optical performance (i.e. low transmittance and high haze) (Supplementary Fig. 9). Finally, to endow the MTPW with water repellency, it was coated with superhydrophobic SiO$_2$ nanoparticles by an ultrasonic spray-coating method. We chose this method for two reasons. First, compared to other coating methods, such as spin-coating and dip-coating, spray-coating can produce large-area thin films with excellent uniformity. Second, the ultrasonic machine can atomize the ethanol solvent of the SiO$_2$ nanoparticle solution, minimizing the damage to the T-Perovskite induced by the -OH group. In addition, the sample stage of the spray-coating machine was set to 50 °C to further accelerate the evaporation of ethanol on the perovskite surface. The MTPW was successfully fabricated without influencing the perovskite underneath. It should be noted that the PHPS layer is indispensable for the holistic structure of the window because it serves four pivotal functions: (1) Preserving T-Perovskite integrity during the SiO$_2$ layer deposition process: without the PHPS protection, the T-Perovskite layer can be easily damaged by the ethanol used in the SiO$_2$ dispersion for the spray coating process (Supplementary Fig. 10a); (2) Enabling the stable hydrophobicity of the MTPW: hydrophobic SiO$_2$ possesses a porous structure, while the perovskite demonstrates a marked ability for water absorption. In the absence of a dense PHPS protective layer, water droplets can easily collapse under the strong adsorptive force and penetrate the perovskite layer through the pores of the SiO$_2$ layer, thus eventually destroying the perovskite (Supplementary Fig. 10b and S10c); (3) Insulating against excess water vapor to further extend the durability of the thermochromic perovskite: the durability of a window solely coated with SiO$_2$ nanoparticles solely coated window falls short of being comparable with the one with PHPS layer (Supplementary Fig. 10d); and (4) Reducing the haze of the window (Supplementary Fig. 10e and more elaborations are in the subsequent section). In the discussion in the following sections, the samples obtained in each fabrication step are named the T-Perovskite window (TPW), PHPS-coated T-Perovskite window (PTPW), and MTPW, as shown in Fig. 1e.

## Optical performance and phase transition properties of the MTPW

Excellent optical performance is always the first consideration in window designs. Considering the multilayer structure of the MTPW, we carefully optimized each layer to maximize the overall optical transparency. Figure 2a shows the transmittance spectra of the MTPW and TPW. The MTPW exhibits a $\tau_{lum}$ of 83.4% in the cold state and 30.4% in the hot state, with a $\Delta\tau_{sol}$ of 24.4%, while the $\tau_{lum}$ of the TPW was 78.2% and 28.8% in the cold and hot states, respectively, with a $\Delta\tau_{sol}$ of 25.4%. Notably, both the $\tau_{lum,cold}$ and $\tau_{lum,hot}$ of the MTPW were higher than those of the TPW because of the decreased reflectance (Supplementary Fig. 11). The key reason for the lower reflectance is less light scattering at the surface causing a reduction in hazing. Optical haze, which can be defined as the ratio of diffuse transmittance to total transmittance and includes both the specular and diffuse parts, is an important parameter for windows (Supplementary Fig. 12)[41,42]. A low-haze window provides a clear view, while a high-haze window obscures the view via light distortion, although the total transmittance remains high. Even though the pristine TPW has advantages in both $\tau_{lum}$ and $\Delta\tau_{sol}$ compared to other solution-based smart windows (e.g., for VO$_2$ thermochromic windows, $\tau_{lum}$ and $\Delta\tau_{sol}$ are typically ~50% and ~10%, respectively[43,44]), the high haze of TPW windows is neglected in most studies. As shown in Fig. 2b, the pavilion 10 m from the TPW cannot be seen, whereas it is clearly visible through the MTPW. Optical measurements also verified that the optical haze of the TPW was as high as 90%, whereas that of the PTPW and the MTPW was markedly reduced to as low as ~20% and ~30%, respectively (Fig. 2c).

To explain this phenomenon, the surface morphology of TPW, PTPW and MTPW was studied and compared in Fig. 2d$_1$ - d$_6$. The rough and uneven surface of the TPW (Fig. 2d$_1$) is caused by an excess of MAI used to achieve thermochromism in the perovskite precursor, which in turn affects the crystallization process[27,28]. The 3D morphology, quantitatively determined by a surface profiler, reveals that the roughness $Ra$ of the TPW surface is 208 nm (Fig. 2d$_2$) (Thickness of T-Perovskite: ~1.6 μm). However, after coating with PHPS, the surface of the PTPW becomes quite flat, as shown in Fig. 2d$_3$, and the $Ra$ is significantly reduced to 55 nm (Fig. 2d$_4$) (Thickness of PHPS: ~1.8 μm). Based on this observation, we quantitatively explain why the MTPW has enhanced optical performance: Fig. 2e shows the light path at a T-Perovskite surface. According to the Rayleigh roughness criterium[45], if the surface can meet the following equation, then the surface can be considered as a smooth surface (a detailed derivation is given in Supplementary Text 1).

$$\sqrt{\frac{\sum_{A=1}^{N}(h_A-[h])^2}{N}}=RMS<\frac{\lambda}{4(n_1\cos\alpha_i-n_2\cos\alpha_t)}, \quad (2)$$

where $[h]$ is the mean value of the rough surface height and $\sqrt{\frac{\sum_{A=1}^{N}(h_A-[h])^2}{N}}$ is the root mean square average roughness ($RMS$).

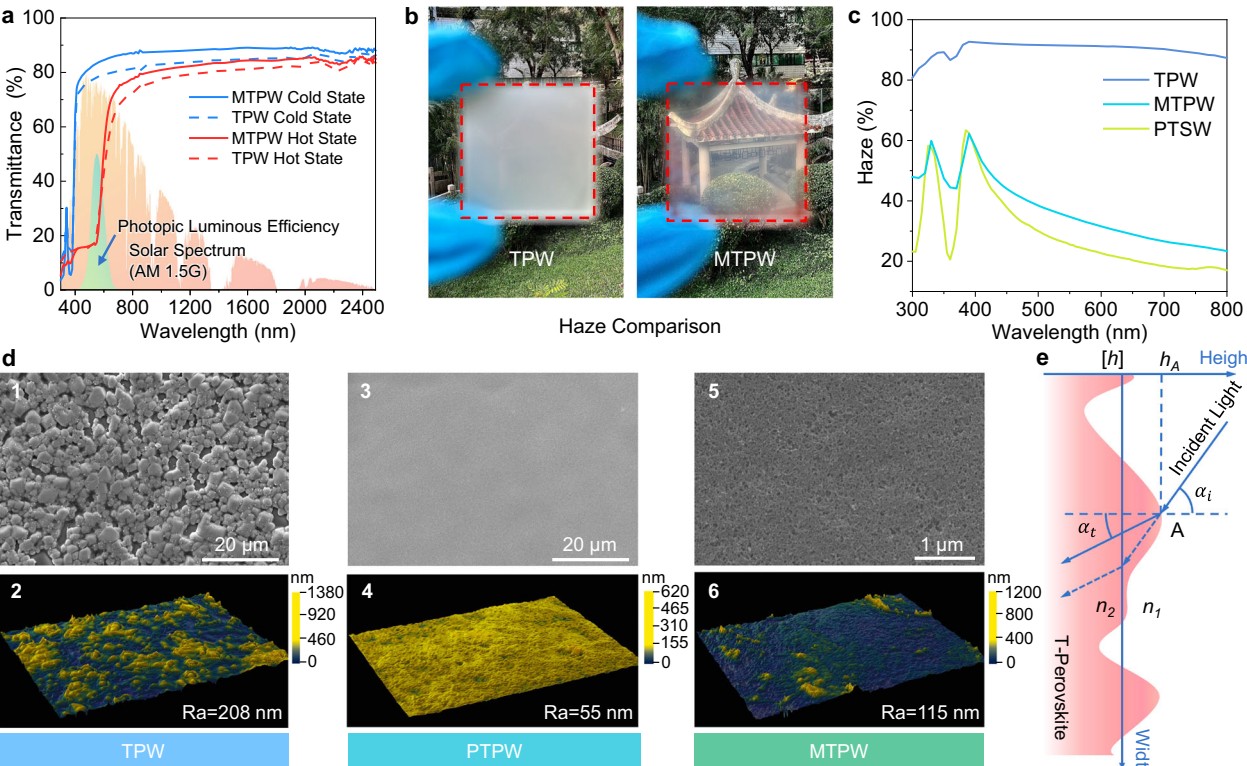

**Fig. 2 | Optical characterization of the mask-inspired thermochromic perovskite window (MTPW). a** Transmittance spectra of the pristine thermochromic perovskite window (TPW) and MTPW, together with the AM1.5 G solar spectrum. **b** Photographs of the TPW and MTPW comparing their optical haze. **c** Optical haze of the TPW, perhydropolysilazane-coated thermochromic perovskite window (PTPW) and MTPW in the wavelength range of 300–800 nm. **d** 1, 3 and 5 are the surface scanning electron microscopy (SEM) images of the pristine TPW, PTPW and MTPW, respectively, while 2, 4 and 6 are the 3D optical surface profiles and roughness of the pristine TPW, PTPW and MTPW, respectively. **e** Light propagation through a rough surface. Source data are provided as a Source Data file.

Equation (2) indicates that when the two media have closer refractive indices, the tolerance to the RMS is larger, and vice versa. Both the rough surface of the TPW and the large refractive index difference between air ($n \sim 1.0$) and the T-Perovskite ($n \sim 2.0$) cause strong scattering when light encounters peaks and valleys, causing a high optical haze and a reduced specular transmittance. PHPS has an $n$ value relatively close to that of the perovskite compared to air. Therefore, the adoption of PHPS on top can improve the light propagation path tolerance to the surface roughness (Eq. 2), reducing the light scattering at the T-Perovskite surface boundary.

To verify the above explanation, a finite-difference time-domain (FDTD) simulation was conducted to analyze light propagation through different surfaces (Supplementary Figs. 13 and 14). The complex refractive indices of PHPS and the T-Perovskite (H-MAPbI$_{3-x}$Cl$_x$) in both the cold and hot states were measured by spectroscopic ellipsometry and results are shown in Fig. 1d. Supplementary Movie 2 demonstrates the simulated light propagation process at the air/window interface of an ideally smooth TPW, a rough TPW and a PTPW. It can be seen that depositing PHPS on the rough TPW surface can be helpful to maintain the original propagation path of the light and effectively suppress the scattering at the T-Perovskite surface compared with that for the rough TPW. In addition, the angular distributions of scattered light in the reflected and transmitted fields for the rough TPW and smooth PTPW were also extracted from the FDTD simulation based on the bidirectional scattering distribution function (BSDF). The BSDF can radiometrically characterize the light scattering at a surface as a function of the angular positions of the incident and scattered beams[46]. The scattered light in both the transmitted and reflected fields for the rough TPW surface is distributed over a wider angular range (Fig. 3a$_1$ – a$_3$), leading to a blurry visual effect. In

contrast, for the PTPW surface, both the transmitted and reflected lights are concentrated in a smaller angular range (Fig. 3b$_1$ – b$_3$), resulting in a clearer view. In short, we conclude that the top PHPS layer significantly enhances the optical performance through a large reduction in the haze and an increase in the total transparency by smoothing the originally rough surface.

The top layer of the MTPW is a superhydrophobic layer of nanoparticles to achieve water repellency. Among widely used nanoparticles, SiO$_2$ has the lowest refractive index in the visible light range of 1.45, so it was selected to minimize light scattering. Moreover, according to the Mie scattering theory, the particle size also influences the scattering efficiency ($Q_s$). We calculated the $Q_s$ of a SiO$_2$ nanosphere as a function of the particle diameter across the solar spectrum using FDTD simulations (Supplementary Fig. 15). Figure 3c shows that $Q_s$ decreases as the diameter decreases. $Q_s$ is very small when the particle diameter is 20 nm, indicating a limited influence on the optical performance. Thus, SiO$_2$ nanoparticles of ~20 nm in diameter were chosen for the fabrication of the MTPW (Supplementary Fig. 16). As confirmed in Supplementary Fig. 17, the transmittance spectrum of the MTPW is almost the same as that of the PTPW without SiO$_2$ nanoparticles. The haze only slightly increases due to an increase in the surface roughness (Fig. 2c, Fig. 2d$_5$ and Fig. 2d$_6$).

For a thermochromic smart window, in addition to its excellent optical performance, phase transition properties, namely a low transition temperature ($T_c$) and a short transition time ($t$) are also desired. A relatively low $T_c$ is one of the advantages of the H-MAPbI$_{3-x}$Cl$_x$ perovskite. As a PHPS layer and SiO$_2$ nanoparticles are added to the T-Perovskite, whether the extra coatings affect the $T_c$ in the heating ($T_{c,h}$) and cooling ($T_{c,c}$) processes must be examined. As shown in Fig. 4a, the temperature-dependent transition processes of the TPW

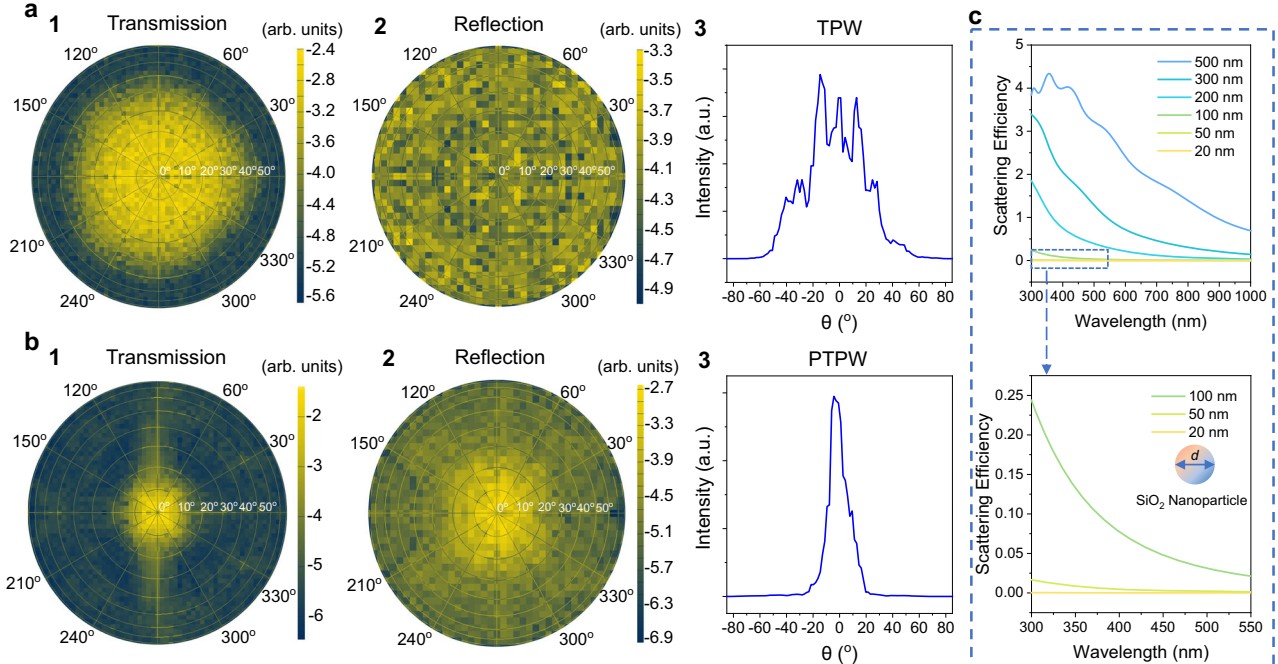

**Fig. 3 | Angular distribution of scattered light. a** 1 and 2 are the angular distributions of scattered light in the transmitted and reflected fields of the pristine thermochromic perovskite window (TPW), respectively. 3 is the transmittance distribution intensity originating from the slice of the transmitted-field radial plot. **b** 1 and 2 are the angular distributions of scattered light in the transmitted and reflected fields of the perhydropolysilazane-coated thermochromic perovskite window (PTPW), respectively. 3 is the transmittance distribution intensity originating from the slice of the transmitted-field radial plot. **c** Scattering efficiency of $SiO_2$ nanoparticles of different sizes. *d* refers to the diameter of the $SiO_2$ nanoparticles. Source data are provided as a Source Data file.

and the MTPW are almost identical. The $T_{c,h}$ and $T_{c,c}$ values of the pristine TPW extracted from the measurements are 43.2 °C and 35.6 °C, respectively, while those of the MTPW are 43.4 °C and 35.9 °C, respectively, implying that the coatings have no influence on the $T_c$. In addition, as shown in Fig. 4b, the pristine TPW completes its color switching in 65 s and 120 s during the heating and cooling processes, respectively. For the MTPW, because the PHPS and $SiO_2$ nanoparticle layers limit the water vapor transport rate, the transition times in the heating and cooling processes are inevitably extended to 120 s and 300 s, respectively, which are still acceptable for practical applications.

**Superhydrophobicity, stability and application of the MTPW**

Figure 4c, d compare the wettability of the pristine TPW and PTPW. The pristine TPW is hydrophilic, with an initial contact angle (CA) of 21.2°, while the T-Perovskite film is immediately damaged by the water droplet (Supplementary Fig. 18). The CA decays to 12.4° in 1 s and cannot be measured by the CA meter after 3 s, indicating severe water corrosion. The PTPW exhibits a higher CA of 98.4°, but the CA rapidly decreases to 54.1° after 9 min. The poor hydrophobicity of both the TPW and PTPW cannot effectively repel water on the window surface. However, the superhydrophobic feature of the T-Perovskite smart window is desired to physically impede harmful moisture and water and prolong the window life. As described above, superhydrophobic $SiO_2$ nanoparticles 20 nm in diameter were coated on the PHPS layer. Supplementary Fig. 19 shows surface SEM images of $SiO_2$ nanoparticles for different numbers of spray-coating cycles. As the number of spray-coating cycles increases, the packing density of $SiO_2$ nanoparticles increases, and the surface becomes almost fully covered by $SiO_2$ nanoparticles when the number of cycles is above 25. Figure 4e demonstrates the influence of the spray-coating cycle number on the CA, in which a rising trend from 115.3° with 5 cycles to 160.8° with 50 cycles is observed. After 50 cycles, the CA is almost unchanged and even slightly decreases. This occurs because the fully covered $SiO_2$ nanoparticles lead to an increase in the area of the liquid–solid contact

interface, which results in a higher surface adhesion force[47]. In addition, the sliding angle of the MTPW is only 5.7° for 50 cycles, which is beneficial for the window. Therefore, the optimal $SiO_2$ spray-coating protocol with 50 cycles was used in this work. After deposition of the $SiO_2$ nanoparticles, the MTPW demonstrates superhydrophobicity, with a stable CA above 160° (Fig. 4c). When a dyed droplet was dripped on the surface of the MTPW, the droplet spread into a compressed disk shape and bounced off the surface, implying prominent superhydrophobicity (Fig. 4f and Supplementary Movie 3). Such an excellent superhydrophobic property enables the MTPW to efficiently repel water and effectively prevent T-Perovskite degradation by bulk water. Furthermore, a water flushing test was conducted to test the waterproof ability of the MTPW. As shown in Supplementary Movie 4, the MTPW was placed under a faucet and flushed with high-speed water (flow rate: ~4.7 m/s). The T-Perovskite survived, and it still can demonstrate reversible color switch ability upon heating and cooling, which confirms the excellent water repellence of the MTPW. In addition, the abrasion test for $SiO_2$ nanoparticles coating shows reliable hydrophobic functions for practical applications (Supplementary Fig. 20).

To inspect the durability enhancement of the MTPW, a comprehensive investigation was carried out. The optical stability of a smart window is the key to evaluate its durability. Therefore, the long-term optical performance of the MTPW was examined via humidity resistance tests at different relative humidity (RH) levels. To precisely control the temperature and humidity, the experiment was conducted in an environmental test chamber under 23 °C and 60% RH (normal ambient environment) and 35 °C and 80% RH (hot and humid environment) conditions. As shown in Fig. 5a, b, at 23 °C and 60% RH, all features, including the $\tau_{lum,cold}$, $\tau_{lum,hot}$ and $\Delta\tau_{sol}$ of the TPW displays a distinct tendency to decay during the aging test due to moisture-induced degradation. In particular, $\tau_{lum,hot}$ significantly increases from 28.8% to 68.8% in 5 days, resulting in a large decrease in $\Delta\tau_{sol}$ from 23.1% to 4.8%, which indicates a loss of thermochromism. In sharp

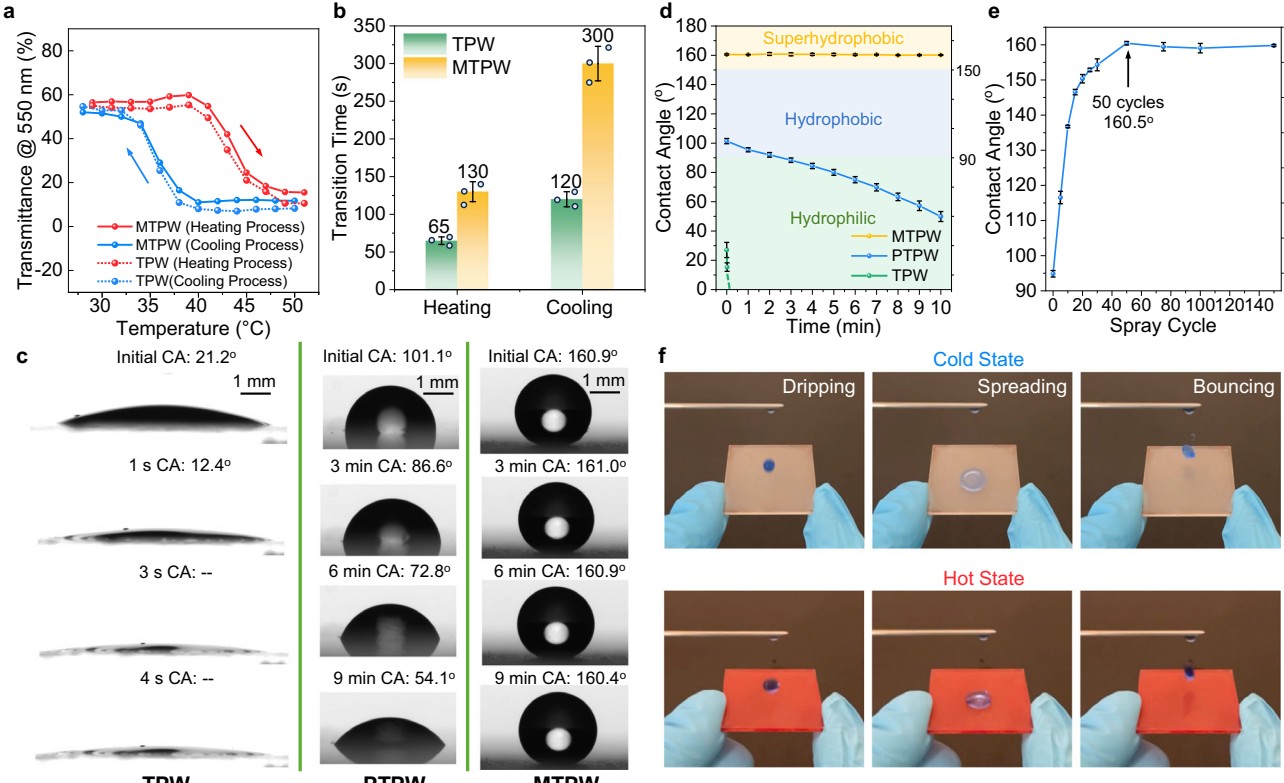

**Fig. 4 | Phase transition properties and superhydrophobicity of the mask-inspired thermochromic perovskite window (MTPW). a** Transmittance of the MTPW and the pristine thermochromic perovskite window (TPW) at 550 nm as a function of temperature showing the transition upon temperature changes. The red arrow indicates the heating process with a temperature increase, while the blue arrow indicates the cooling process with a temperature decrease. **b** Transition times of the MTPW and pristine TPW. **c, d** Time-dependent contact angle measurement of the TPW, perhydropolysilazane-coated thermochromic

perovskite window (PTPW), and MTPW. The yellow, blue and green regions in Fig. 4d represent the superhydrophobic, hydrophobic and hydrophilic areas, respectively. **e** Water contact angle of the MTPW *vs.* spray-coating cycle for the SiO$_2$ nanoparticles during the fabrication process. **f** Demonstration of the super-hydrophobicity of the MTPW. A dripped water droplet bounces off from the MTPW in both the cold and hot states. The error bars in (**b**), (**d**) and (**e**) represent the standard deviations from three parallel measurements. Source data are provided as a Source Data file.

contrast, the MTPW maintains its optical performance over 40 days, with its $\Delta\tau_{sol}$ remaining above 20% at the 45th day of the aging test. The significantly enhanced durability can be ascribed to the strong protection of the PHPS and SiO$_2$ superhydrophobic layers, which insulate the T-Perovskite film from a continuous penetration of abundant water vapor. Similar durability extension was also observed under hot and humid environments (Supplementary Fig. 21a, b). By comparing the decay rates of $\Delta\tau_{sol}$ (Fig. 5c and Supplementary Fig. 21c), the lifespan of the MTPW is found to be considerably extended compared with that of TPW by 37 times in the normal ambient environment and by 98 times in the humid environment, making it reliable for practical applications.

Because of the water vulnerability of T-Perovskite films, the issue of Pb leakage has attracted considerable attention. In the thermochromic process, a T-Perovskite undergoes a water-induced phase transformation from MAPbX$_3$ at the hot state to MA$_4$PbX$_6$ 2H$_2$O in the cold state, with a reduction in dimensions. The formed low-dimensional dihydrated perovskite is further decomposed as shown in equation 3:

$$MA_4PbX_6 \cdot 2H_2O \longrightarrow PbX_2 + 4MA^+ + 4X^- + 2H_2O \qquad (3)$$

The presence of strong hydrogen bonding between water molecules and the toxic PbX$_2$ leads to a high solubility constant ($K_{sp} = \sim 10^{-8}$) of PbX$_2$ in water[21], which causes severe Pb leakage, threatening the environment and human health. To examine the Pb leakage issue, both the pristine TPW and MTPW were immersed in deionized water. As shown in Fig. 5d, the pristine TPW immediately changed color to

yellow within 1 s, implying Pb leakage. The X-ray diffraction (XRD) pattern of the yellow film shown in Supplementary Fig. 22 proves that the T-Perovskite film decomposed to PbI$_2$[48]. In contrast, the MTPW maintains the transparent state for as long as 5 min underwater. Moreover, the color still changed upon heating, indicating that the T-Perovskite is well preserved and that the thermochromic effect is maintained. To further estimate the amount of Pb$^{2+}$ ions in the contaminated water, inductively coupled plasma–mass spectrometry (ICP–MS) measurements were conducted, as shown in Fig. 5e. The Pb$^{2+}$ concentration in water for the pristine TPW reaches 4.023 mg/L in 30 min, which was 66 times higher than that for the MTPW (0.061 mg/L). These results strongly suggest that the superhydrophobic MTPW is capable of immobilizing Pb$^{2+}$ and reducing Pb leakage, making the MTPW more eco-friendly.

The MTPW exhibits better moisture and water resistance and simultaneously less lead leakage compared with the pristine TPW. Moreover, the all-solution-based fabrication process without the need for any extra toxic antisolvent provides the scalability of the MTPW. These advantages enable the direct fabrication of a flexible MTPW window film without the need for sealing, which can easily be integrated into existing glass windows. Here, an MTPW film with high flexibility was fabricated by coating the T-Perovskite, PHPS, and SiO$_2$ nanoparticles in sequence on a PET film (Fig. 5f). Compared with widely used tinted window films that have a dark color and can block part of visible and near-infrared light to constantly reduce indoor solar heat gain even in cold weather, the solar transmittance of our MTPW film is dynamically modulated with temperature changes, which maintains

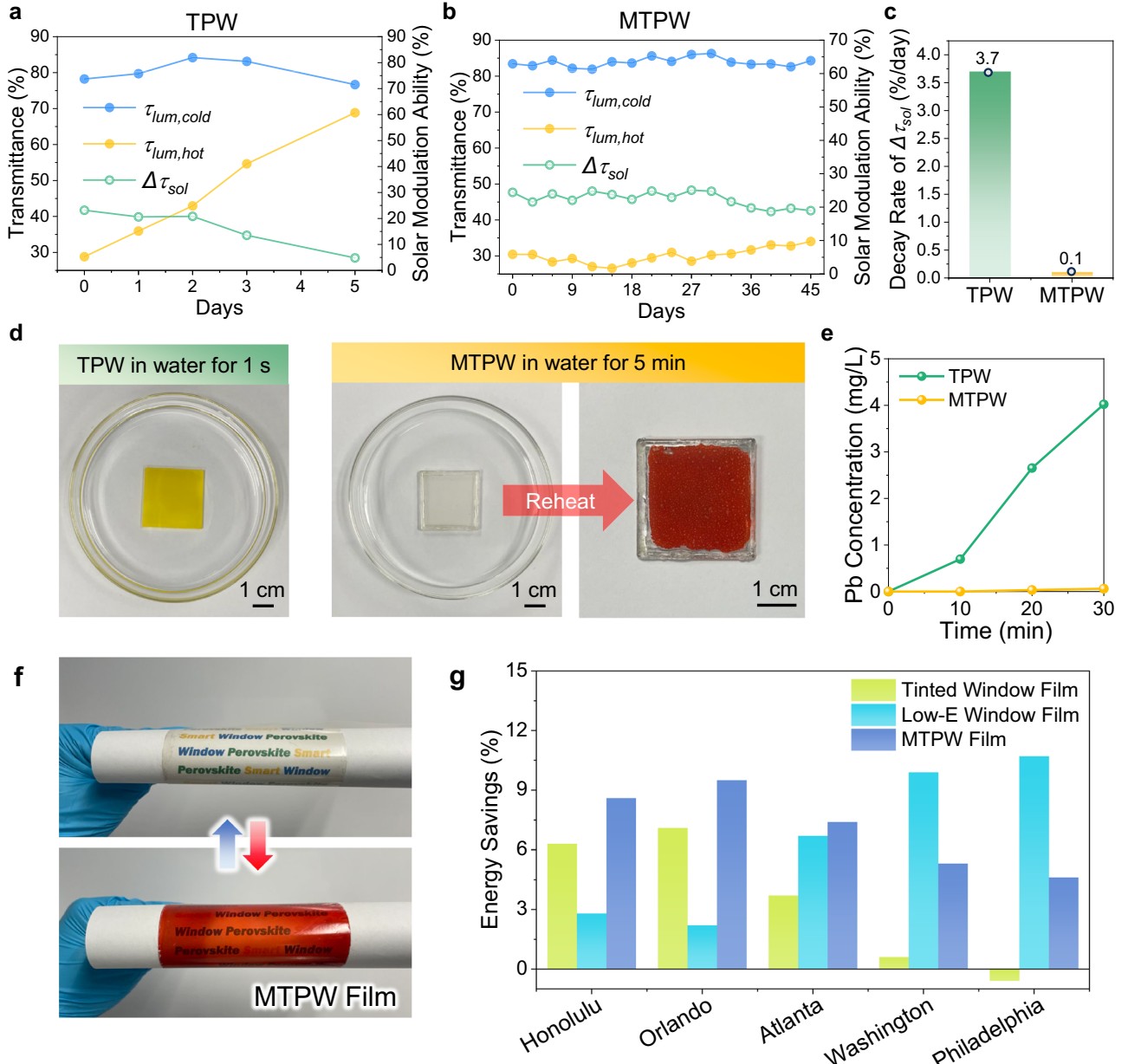

**Fig. 5 | Durability test and application of the mask-inspired thermochromic perovskite window (MTPW). a**, **b** Optical performance ($\tau_{lum,hot}$, $\tau_{lum,cold}$ and $\Delta\tau_{sol}$) of the pristine thermochromic perovskite window (TPW) and the MTPW in the ambient environment. **c** Decay rate of $\Delta\tau_{sol}$ for the TPW and MTPW in the ambient environment. **d** Water repellence ability test for the pristine TPW and MTPW. **e** Pb concentration after soaking in water for the pristine TPW and MTPW. **f** Photographs of the flexible MTPW film in the cold and hot states. Red and blue arrows indicate heating and cooling processes, respectively. **g** Energy savings over a year obtained by using the MTPW, the tinted window film and the Low-e window film compared with a normal window. The energy consumption of the building in Energyplus simulation was collected, and the energy savings were calculated as the difference of energy consumption with normal windows and with other windows, divided by the energy consumption with normal windows. Source data are provided as a Source Data file.

the useful solar heat gain in cold weather/seasons but reduces undesirable solar heat gain in hot weather/seasons (Supplementary Fig. 23a, c). Therefore, the MTPW is expected to have better all-year energy-saving ability. EnergyPlus modeling (Supplementary Text 2 and Supplementary Fig. 24) was performed on a commercial building (Supplementary Table 1) using different windows (Supplementary Table 2) in typical cities in the U.S (Supplementary Table 3). The results verify that the use of the MTPW film shows higher energy-saving potential than the tinted window film, especially in the northern areas where buildings also have a high heating demand in the winter (Fig. 5g). The MTPW film was also compared with Low-E window films (Supplementary Fig. 23b) that have a low emissivity ($\varepsilon = 0.109$) in the

long wave infrared region (Supplementary Fig. 23d). Due to the high thermal radiation reflectance, using Low-E film can achieve better thermal insulation for the indoor environment thus leading to even higher energy-saving performance than the MTPW film in those areas that have long and cold winters (Fig. 5g, Supplementary Fig. 24c, d). While, in hot areas or the summer time, the MTPW film demonstrates better energy-saving potential than the Low-E film (Fig. 5g, Supplementary Fig. 24e, f), since the high thermal reflection of the Low-E film blocks the heat dissipation. These results suggest that the MTPW window film is promising for reducing HVAC energy consumption throughout the year, and it should be noted that due to the light modulation region of the T-Perovskite being in the visible light range, it

can also be conveniently integrated with a Low-E coating (i.e. functional region in the IR range) to develop more advanced multi-layer window films to achieve higher energy saving abilities[16].

## Discussion

In this study, inspired by the structure of a mask, a TPW with improved optical properties and enhanced durability was explored. The unique three-layer structure enabled the MTPW to limit water vapor transmission in the right amount to trigger thermochromism and prevent excess-humidity-induced degradation of the T-Perovskite. Simultaneously, benefitting from the superhydrophobic surface (CA of 160°), the MTPW achieved an excellent waterproof ability and significantly suppressed lead leakage by 66 times compared with the pristine TPW when submerged in water. Most importantly, the $\Delta\tau_{sol}$ of the MTPW remained above 20% for 45 days in the ambient environment, with a decay rate reduced by 37 times compared with that of the TPW. This result is the longest record reported for T-Perovskite smart windows to date. Additionally, the MTPW demonstrated a reversible color change with $\tau_{lum}$ of 83.4% and 30.4% in the cold and hot states, respectively, a high $\Delta\tau_{sol}$ of 24.4%, a low $T_c$ and a short $t$ of <5 min. In particular, the middle PHPS layer in the MTPW acted as an optical buffer, leading to a low optical haze of 30%, thus making the T-Perovskite window clear. Finally, we fabricated a cost-effective flexible MTPW film with desirable thermochromic properties. EnergyPlus modeling confirmed that its smart optical switching function can lead to significant all-year energy savings. In the future, the degradation mechanism of the T-Perovskite needs to be further investigated to better balance the thermochromism and decomposition reaction, which may purposely guide the protection layer design. In addition, there is still potential to further reduce the optical haze of the MTPW. For instance, exploring encapsulation materials with a refractive index that matches T-Perovskite could further decrease RMS tolerance and subsequently reduce the haze. Another potential method is to utilize the anti-solvent crystallization technique to produce smoother T-Perovskite films. However, it is important to consider the toxicity of the anti-solvent and scalability issues associated with this approach. Moreover, the current T-Perovskites exhibit optical modulation primarily in the visible light region. To enhance the solar modulation ability, it would be worthwhile to investigate new broadband-modulated perovskite materials that span from the visible light to NIR range. Possible approaches include tuning the bandgap of T-Perovskite or integrating T-Perovskite with other materials like $VO_2$. This work provides an easy yet effective strategy to realize T-Perovskite windows that are simultaneously environmentally stable, optically clear and eco-friendly through multilayer surface design, further making the practical application of T-Perovskite smart windows in energy-efficient buildings possible.

## Methods

### Materials and chemicals

$CH_3NH_3I$ (MAI, 99.5%) was provided by Xi'an Polymer Light Technology. $PbCl_2$ (99%) was purchased from Sigma–Aldrich. Dimethylformamide (DMF, ≥99.5%) was purchased from Alfa Aesar. PHPS was supplied by Iota Silicone Oil. The fluorinated $SiO_2$ nanoparticle/ethanol solution was provided by Solmont Tech.

### Fabrication of the MTPW

Glass substrates were cleaned with detergent, ethanol, and deionized (DI) water in an ultrasonic bath for 15 min, followed by drying with $N_2$. The glass substrates were further cleaned in a plasma cleaner (FARI GD-5) for 200 s. The H-$MAPbI_{3-x}Cl_x$ precursor was synthesized by mixing MAI and $PbCl_2$ at a molar ratio of 6.5:1 in DMF solvent, followed by stirring at 50 °C for 1 h. The H-$MAPbI_{3-x}Cl_x$ precursor was spin-coated by a spin coater (Laurell H6-23) at 2000 rpm for 15 s, followed by annealing at 100 °C for above 1 h to evaporate the residual DMF. Then, 20 wt% PHPS in dibutyl ether solution was spin-coated on the

T-Perovskite layer at 500 rpm for 15 s, and the sample was cured in air at 100 °C for 3 h. Then, the sample was moved to an ultrasonic spray-coating machine (UC 330, Siansonic Technology) to coat the $SiO_2$ nanoparticles. The detailed parameter settings for the spray-coating machine are shown in Supplementary Text 3.

### Characterization

The FTIR spectrum of PHPS was obtained by a PerkinElmer Spectrum 3. The SEM images and energy-dispersive X-ray spectroscopy (EDS) mappings of the samples were obtained by a FEI Quanta 450. Transmission electron microscopy (TEM, 2010F, Jeol) was used to characterize the size of the $SiO_2$ nanoparticles. The roughness of the samples was measured by 3D surface metrology (Bruker NPFLEX).

The transmittance spectra were obtained by a UV–Vis-NIR spectrophotometer from 300 nm to 2500 nm (Lambda 1050, Perkin Elmer equipped with a 150 mm integrating sphere detector). A tailor-made temperature controller (including a heater, a T-type thermocouple, and a temperature controller) was attached to the window sample to control the temperature while measuring the transmission in both the cold (25 °C) and hot (60 °C) states. The luminous transmittance ($\tau_{lum}$) of each window was calculated based on CIE (International Commission on Illumination) 1931 standard observer by $\tau_{lum} = \frac{\int_{\lambda=380nm}^{780nm} \bar{y}(\lambda)\tau(\lambda)d\lambda}{\int_{\lambda=380nm}^{780nm} \bar{y}(\lambda)d\lambda}$ to quantify the amount of transmitted visible light, where $\tau(\lambda)$ is the transmittance of the window at wavelength $\lambda$. $\bar{y}(\lambda)$ is the photopic luminous efficiency of the human eye. The total solar transmittance is defined as transmittance $\tau_{sol}$ ($\tau_{sol} = \frac{\int_{\lambda=300nm}^{2500nm} AM_{1.5}(\lambda)\tau(\lambda)d\lambda}{\int_{\lambda=300nm}^{2500nm} AM_{1.5}(\lambda)d\lambda}$, where $AM_{1.5}(\lambda)$ is the AM1.5 G solar irradiance spectrum). $\Delta\tau_{sol}$ was calculated by $\tau_{sol} = \tau_{sol}^{cold} - \tau_{sol}^{hot}$. Following ASTM D1003 "Standard Method for Haze and Luminous Transmittance of Transparent Plastics", the haze was calculated as haze $= \left(\frac{T_4}{T_2} - \frac{T_3}{T_1}\right) \times 100\%$, where $T_1$ is the incident light, $T_2$ is the total light transmitted by the sample, $T_3$ is the light scattered by the equipment, and $T_4$ is the light scattered by the sample and equipment.

### Transition properties measurements

To measure the transition temperature $T_c$ of the T-Perovskite, a heating and cooling cycle was conducted for the samples on a high-precision temperature-controlled hot plate (CHEMAT 4AH) between room temperature and 60 °C at intervals of 2 °C. For each temperature set point, the samples were kept on the hot plate for 5 min to ensure the stability of the color. At the same time, the transmittance at 550 nm was measured by a Lens Transmission meter (SDR8508). Then, $T_c$ was calculated by plotting the first derivative of the transmittance with respect to the temperature as a function of temperature, where $T_c$ is the minimum value point of the first derivative. The transition time was measured by observing the tinting and color fading of smart windows in the heating and cooling processes at the corresponding transition temperature point.

### Contact angle measurements and durability test

The water CA and sliding angle measurements were performed by means of a static CA meter (Biolin Theta), and 4 μL water droplets were dripped on each test surface. For the abrasion test, the coated side of the $SiO_2$ nanoparticles was placed face down on a sheet of 1500 grit sandpaper. Under a weight of 100 g, the window sample was longitudinally and transversely abraded for a distance of 10 cm respectively, which constituted one cycle of abrasion. To conduct the durability test, the pristine TPW and MTPW were placed in an environmental test chamber, and the conditions of the environmental test chamber were set to 23 °C/60% RH and 35 °C/80% RH to mimic the ambient environment and a hot and humid environment, respectively.

The sample was heated and cooled once a day. In the hot condition, the sample was kept at the transition point to mimic its colored state in realistic scenarios. The transmittance of the sample in the cold and hot states was measured to judge the stability of the T-Perovskite. For the Pb leakage test, the Pb concentration in the contaminated water was detected by an ICP–MS instrument (Perkin Elmer 2000).

### FDTD simulations
To conduct the FDTD simulation, the refractive indices of PHPS and H·MAPbI$_{3-x}$Cl$_x$ in the cold and hot states were measured by an ellipsometer (J.A. Woollam RC2). In the FDTD simulation (Lumerical software), to obtain the reflection and transmission spectra of the window in the wavelength range of 0.3–2.5 μm, a plane wave was placed on top as the light source. The rough H·MAPbI$_{3-x}$Cl$_x$ and PHPS layers were created based on the roughness measured by 3D surface metrology. The boundary conditions along the z-direction were defined as perfectly matched layers (PMLs), while symmetric boundary conditions were applied for the x- and y-directions. To verify the accuracy of the FDTD model, the simulated transmittances of the TPW and PTPW were compared with the experimental results, and their similar transmittance spectra, as shown in Supplementary Fig. 14, prove the reliability of the FDTD model. Additionally, the scattering cross-section of a single SiO$_2$ nanoparticle was simulated to calculate its scattering efficiency. A total-field scattered field (TFSF) source was utilized as the incident light, and the boundary conditions along the x-, y-, and z-directions were all defined as PMLs. The mesh sizes were set as 0.5 nm. The output from the FDTD simulation was the scattering cross-section ($C_{sca}$) of a spherical particle, and the scattering efficiency coefficient ($Q_{sca}$) was $C_{sca}$ normalized as $Q_{sca} = \frac{C_{sca}}{\pi r^2}$, where πr$^2$ is the geometrical cross-sectional area of the scattering particle.

### Reporting summary
Further information on research design is available in the Nature Portfolio Reporting Summary linked to this article.

## Data availability
All data are available in the main text or the supplementary information. Source data are provided with this paper.

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

## Acknowledgements

C.Y.T. acknowledges financial support from the Hong Kong Research Grant Council via General Research Fund (GRF) account of 11200121 and Innovation and Technology Commission via Innovation and Technology Fund (ITF) account of ITS/041/21.

## Author contributions

S.L., Y.L., C.Y.T. and B.L.H. proposed the concept and designed the research; S.L., Y.L., Y.W. and Y.W.D. conducted the experiment and performed the analysis; S.L. finished the original draft of the manuscript; S.L., Y.L., K.M.Y., H.-L.Y., A.K.Y.J. and C.Y.T reviewed and edited the manuscript. C.Y.T. and B.L.H. supervised the project. S.L. and Y.L. contributed equally to this work.

## Competing interests

The authors declare no competing interests.
