## [Peer Review File · Nature Communications]

Mask-Inspired Moisture-transmitting and Durable Thermochromic Perovskite Smart WindowsREVIEWER COMMENTS

Reviewer #1 (Remarks to the Author):

Liu and coworkers have developed a new mask-inspired architecture for perovskite-based thermochromic windows. By making the films more durable, they are directly addressing one of the largest issues with the technology. I find the “mask” story a little bit of a stretch, but it adds a nice narrative to the paper. The work is interesting and may deserve publication in the Nature Communication, as it demonstrates an interesting method to improve durability that may be applicable to future similar technologies. However, we ask the authors to address the following concerns to be reevaluated:

1) Growing the polymeric Si-O-N material film on top of a perovskite film, especially an MA-based one, is a notable achievement. However, it is unclear why the authors state a barrier layer is needed. It only increases the kinetics of switching, which results in poorer performance. They claim it “Insulates against excess water vapor,” but they do not show that this is important for performance. From what I can glean, the barrier layer is simply needed to protect the perovskite layer from the following spray-coating process to deposit the hydrophobic SiO₂ layer, which will likely irreversibly damage the perovskite. This is not clear in the manuscript. The purpose of the barrier layer should be clear, and experiment of depositing the SiO₂ nanoparticles directly on the perovskite should be shown. If the perovskite does survive the nanoparticle, how well does it perform in durability studies?

2) They spend a lot of time discussing the optical advantage of the PHPS film by decreasing haze. This is an interesting result, but the SiO₂ nanoparticle film increases the haze by a significant amount after it is deposited (even after reducing the particle size), which may render the PHPS films ineffective. Is the haze lower or higher when the SiO₂ nanoparticle film is deposited directly on the perovskite surface? Is this not possible for the reason I describe above? If so, the main role, again, is to be a barrier layer for the following deposition. This should also be clear in the manuscript.

I enjoy a good story when reading a paper, but it appears the authors may have gotten tied up in mask storytelling when it comes to the second layer in their mask analog if it is indeed deposited to protect from the subsequent deposition.

3) The window film idea is an interesting one. Building retrofits will be important for reducing energy consumption in buildings. However, they chose a tinted window film with no low-e properties (IR reflection). It is likely not a good comparison for energy savings since low-e films should perform better than the commercial film they chose. How does the thermochromic perovskite film compare to a low-e retrofit film? This should also be addressed in the manuscript to not mislead the readers of the thermochromic film benefits over existing state-of-the-art technology.

Typo on line 78 “hinters” should be “hinders”

Reviewer #2 (Remarks to the Author):

Dear authors,

I would like to commend you on an impressive manuscript. I found it to be clear and quite extensive in terms of the amount of work which has been performed to address the various aspects of T-perovskites for smart window applications. There are, however, a few points which I believe should be addressed prior to publication:

- The improvements which you demonstrate are important and relevant for a potential application of such coatings for smart windows. However, I would like you to critically address some of the key characteristics of the solution which you propose. For one, you mention the importance of the aesthetic aspect of windows.

- o The present solution displays a reddish color in the high temperature state. Red is typically not an attractive color in the architectural glass industry. Can anything be done to modify the color?

- o Although the level of haze has been significantly decreased, 30% is still non-negligible.

- Other window considerations:

- o Water vapor is key to induce the TC effect. Can you comment on the minimum level of relative humidity for this type of window to work? In fact, what would happen if it was indeed encapsulated in a double-glazed window where the water vapor would be minimal.

- o Yes, the presented T-Perovskite has a high $\Delta\tau_{sol}$, but most of the optical change is actually happening in the visible. Can you comment on this? Can the change be increased in the NIR?

- o Can you comment on the mechanical durability of the coating considering the top nanoparticle-based film?

- In the introduction, you mention: light-absorbing phase to a transparent phase as the temperature changes, which results in a color switch. Please clarify that the transparent phase is in the low temperature state.

- Bottlenecks: Points III and IV are very similar: high haze vs optical transparency. Could Point IV be the leakage of lead?

- You indicate that the PHOS consists in a $\text{SiO}_x/\text{SiON}_x$ film; do you expect the refractive index to be slightly higher than 1.5 in this case? 1.55 as indicated further on?

- Have you tested thicker PHPS thicknesses than 1.8 μm ? Was the transition time judged too slow?

- You have quantified the level of roughness of your T-Perovskite films at 208 nm. Fig.1 E shows a geometrical optics representation for the presence of haze which can be misleading; indeed, one is most probably in the Mie scattering regime for visible wavelengths.

- You indicate: "Notably, both the $\tau_{lum,cold}$ and $\tau_{lum,hot}$ of the MTPW were higher than those of the TPW because less light scattering occurred at the surface, and therefore, the amount of haze decreased." This is most probably the result of a decrease in the reflectance, which can easily be measured on an integrating sphere. Note that the level of scattering can also impact the level of absorption.

- Equation (2) indicates that when the two media have closer refractive indices, the tolerance to the RMS is larger. As such, have you considered to encapsulate the T-Perovskite layer within a higher refractive index material (e.g.: SiN which would be index matching)

- "Moreover, the n value of PHPS is between those of air and the T-Perovskite, and it can also serve as an antireflection layer to improve the total transmittance." In order for the film to be antireflective, a specific thickness would be required. The drop in reflection after the addition of the PHSP is most probably due to the lower amount of haze more than to an antireflective effect which would entail interference phenomena. Did the transmission or reflection change with the different thicknesses of PHSP?

- The FDTD study is interesting but the results are expected (especially for Figure 3A). As a result, I feel that the section on the reduction in haze is lengthy and could be shortened. The BSDF data is interesting, however.

- You mention a commercial energy-saving window film; this is quite vague. Why was this particular film chosen?

- Outstanding optical properties is a bit strong. In comparison to traditional T-perovskites perhaps, but in comparison to the overall demands for window coatings, there is still room for improvement.

- Please specify the observer (CIE 1931 or 1964) for the luminance transmittance calculations.

- The manuscript is well written, but I have found a few instances where it could be improved:

o In the abstract: Inspire by the medical mask structure. I would say inspired by the structure of medical masks.

o Hinters should be hindere.

o At both the cold and hot states should be in both the cold and hot states.

o "medical masks have become important personal belongings"; please rephrase.

o to block most of the virus"es"

o we select"ed" a widely used inorganic oxide

o SiO₂ coating normally relies on; SiO₂ coating"s are typically deposited"...

o 100 °C for 1 hour in "a" glove box to form a 1.6 μm-thick T-Perovskite film

o Figure 1: Repels water droplet "s".

o Rayleigh roughness criteri"um"

o Conclusion (not discussion)

o Enthalpy testing chamber? Are you sure enthalpy is the right term here?

Thank you and best regards

Research Article, No. NCOMMS-23-15053

Title: Mask-Inspired Breathable and Durable Thermochromic Perovskite Smart Windows

Response Statements

Dear Editor and Reviewers,

I would like to express my great gratitude to the editor and reviewers for spending time to review this manuscript. All the comments are highly valuable for us to improve our work. Our responses to all the comments from reviewers follow on a point-by-point basis. The relevant revisions in the manuscript are highlighted in yellow. Thank you so much.

Reviewer #1:

Liu and coworkers have developed a new mask-inspired architecture for perovskite-based thermochromic windows. By making the films more durable, they are directly addressing one of the largest issues with the technology. I find the “mask” story a little bit of a stretch, but it adds a nice narrative to the paper. The work is interesting and may deserve publication in the Nature Communication, as it demonstrates an interesting method to improve durability that may be applicable to future similar technologies. However, we ask the authors to address the following concerns to be reevaluated:

Response: Thank you sincerely for your positive evaluation and constructive comments. Your insightful feedback has greatly inspired us to improve our work. We have thoroughly revised the manuscript in accordance with your valuable suggestions. Please find our response statement below.

1. Growing the polymeric Si-O-N material film on top of a perovskite film, especially an MA-based one, is a notable achievement. However, it is unclear why the authors state a barrier layer is needed. It only increases the kinetics of switching, which results in poorer performance. They claim it “Insulates against excess water vapor,” but they do not show that this is important for performance. From what I can glean, the barrier layer is simply needed to protect the perovskite layer from the following spray-coating process to deposit the hydrophobic SiO₂ layer, which will likely irreversibly damage the perovskite. This is not clear in the manuscript. The purpose of the barrier layer should be clear, and

experiment of depositing the SiO₂ nanoparticles directly on the perovskite should be shown. If the perovskite does survive the nanoparticle, how well does it perform in durability studies?

Response: Thank you so much for your great comments. We deeply regret any confusion caused by the unclear description of the PHPS layer's function in the manuscript. In fact, the PHPS layer serves four key functions: (1) Protecting the thermochromic perovskite during the SiO₂ deposition process; (2) Enabling the SiO₂ to achieve stable hydrophobicity; (3) Acting as an insulator to minimize the ingress of water vapor, thereby enhancing the durability of the thermochromic perovskite; and (4) Mitigating the window's haze (discussed in the comment #2 below). In response to the reviewer's suggestions, we tried to directly spray-coat the SiO₂ nanoparticles on the thermochromic perovskite surface. Fig. R1-1 shows the photos of the SiO₂ particles coated thermochromic perovskite window without the PHPS layer, termed as STPW. Notably, the thermochromic effect is still observable in the STPW. However, the absence of the PHPS protective layer results in the appearance of black dots on some regions of the perovskite surface. These black dots are caused by the reaction between the ethanol solvent for the SiO₂ nanoparticle and perovskite. Regrettably, this has a detrimental effect on the optical performance of the STPW, the τ_{lum} is only 70.3% and 37.2% at the cold and hot states respectively. Additionally, the $\Delta\tau_{sol}$ is 13.7%, considerably worse than the pristine thermochromic perovskite window (TPW) and mask-inspired thermochromic perovskite window (MTPW) whose $\Delta\tau_{sol}$ are both higher than 20%.

Furthermore, we examined the hydrophobicity of the STPW. Fig. R1-2 shows that the contact angle (CA) of the STPW is 150.5° at the initial state, but the water droplet collapses in approximately 3.6 s, causing the CA to rapidly decrease to 34.5°, indicating the loss of hydrophobicity. This arises from the fact that the hydrophobic SiO₂ layer possesses a porous structure, while the perovskite demonstrates a marked capability for water absorption. In the absence of a dense PHPS protective layer, the water droplet can easily collapse under the strong adsorptive force and penetrate the perovskite layer through the pores of the SiO₂ layer, thus eventually destroying the perovskite (Fig. R1-3).

In addition, we also tested the durability of the STPW in the ambient environmental condition and the results are shown in Fig. R1-4. Compared with the TPW, the durability of the STPW indeed was prolonged because of the water-repellence ability of SiO₂ particles. However, when compared with the MTPW, the STPW's durability still falls short of being comparable, mainly

due to the augmented protection provided by the additional encapsulation of PHPS.

In summary, from the above experiments, it can be concluded that PHPS not only acts as the barrier layer in the fabrication process, but also is the key to maintain the stable superhydrophobicity and heightened durability. The relevant discussions have been added to the revised manuscript, and it is also provided in Italics below the comment #2 for convenience. Figures are included in the supplementary materials as Fig. S10. Thank you so much for the comment.

Fig. R1-1. Photos of the STPW in the cold and hot states.

Fig. R1-2. The contact angle of the STPW.

Fig. R1-3. The STPW was damaged by the water droplet.

Fig. R1-4. Durability test in the ambient environmental condition (23 °C, RH = 60%).

- They spend a lot time discussion the optical advantage of the PHPS film by decreasing haze. This is an interesting result, but the SiO₂ nanoparticle film increases the haze by a significant amount after it is deposited (even after reducing the particle size), which may render the PHPS films ineffective. Is the haze lower or higher when the SiO₂ nanoparticle film is deposited directly on the perovskite surface? Is this not possible for the reason I describe above? If so, the main role, again, is to be a barrier layer for the following deposition. This should also be clear in the manuscript.

I enjoy a good story when reading a paper, but it appears the authors may have gotten tied up in mask storytelling when it comes to the second layer in their mask analog if it is indeed deposited to protect from the subsequent deposition.

Response: Thank you so much for your comments. Following the experiments for the comment #1 above, we also measured the haze of the STPW as shown in Fig. R1-5. Evidently, compared to the pristine thermochromic perovskite (TPW), the STPW with a SiO₂ nanoparticle layer does not yield notable reduction in haze, whereas both the PTPW and MTPW show much lower haze because the rough surface of thermochromic perovskite is significantly smoothed after coating with the PHPS as discussed in the manuscript. This observation underscores the indispensability of the PHPS layer in the holistic structural design from the view of low haze.

In an overview of the comments #1 and #2, the PHPS layer serves four pivotal functions: (1) Protecting the thermochromic perovskite during the SiO₂ deposition process; (2) Enabling the SiO₂ to achieve stable hydrophobicity; (3) Insulating excess water vapor to further extend the durability of thermochromic perovskite; and (4) Reducing the haze of the window. The relevant discussions of the comments #1 and #2 have been added to the revised manuscript, figures are

included in the supplementary materials as Fig. S10, and it is also provided in Italics below for convenience. Thank you very much for the comment.

Fig. R1-5. The haze comparison of TPW, STPW, MTPW and PWSW.

Highlighted in yellow at the Design and Fabrication of the MTPW:

It should be noted that the PHPS layer is indispensable for the holistic structure of the window because it serves four pivotal functions: (1) Preserving T-Perovskite integrity during the SiO₂ layer deposition process: without the PHPS protection, the T-Perovskite layer can be easily damaged by the ethanol used in the SiO₂ dispersion for the spray coating process (Supplementary Fig. S10A); (2) Enabling the stable hydrophobicity of the MTPW: hydrophobic SiO₂ possesses a porous structure, while the perovskite demonstrates a marked ability for water absorption. In the absence of a dense PHPS protective layer, water droplets can easily collapse under the strong adsorptive force and penetrate the perovskite layer through the pores of the SiO₂ layer, thus eventually destroying the perovskite (Supplementary Fig. S10B and S10C); (3) Insulating against excess water vapor to further extend the durability of the thermochromic perovskite: the durability of a window solely coated with SiO₂ nanoparticles falls short of being comparable with the one with PHPS layer (Supplementary Fig. S10D); and (4) Reducing the haze of the window (Supplementary Fig. S10E and more elaborations are in the subsequent section).

3. The window film idea is an interesting one. Building retrofits will be important for reducing energy consumption in buildings. However, they chose a tinted window film with no low-e properties (IR reflection). It is likely not a good comparison for energy savings

since low-e films should perform better than the commercial film they chose. How does the thermochromic perovskite film compare to a low-e retrofit film? This should also be addressed in the manuscript to not mislead the readers of the thermochromic film benefits over existing state-of-the-art technology.

Response: Thank you so much for your comments. The thermochromic functional region of the MTPW film is mainly in the visible light range, so we chose a visibly-tinted window film (Fig. R1-6A) in the building energy simulation to demonstrate the strength of the color switch film for the all-year energy savings. However, we agree with the reviewer that it is necessary to conduct the energy-saving simulation more comprehensively to avoid misleading the readers that the thermochromic film benefits all window technologies. Therefore, we added the Low-E window film in the simulation for comparison. It should be noted that different from the MTPW film and the tinted window film, Low-E window film focuses on reducing heat transfer through low emittance (i.e high IR reflection). Therefore, Low-E film is especially suitable for cold areas to mitigate heat loss (i.e. thermal radiation from indoor to outdoor) in winter.

As shown in Fig. R1-6B, we chose the commercial Low-E window film ($\text{SnO}_2\text{-Ag-SnO}_2$ coating structure) that has high luminous transmittance (84%, Fig. R1-6C) but a low emissivity of 0.109 in the mid infrared region (Fig. R1-6D), which is much lower than tinted and MTPW window films whose emissivity are larger than 0.9. The simulation in five cities of the U.S. (Fig. R1-8A) shows that Low-E window film has a better energy-saving potential than the MTPW and tinted film in those areas that have long and cold winters, such as Washington and Philadelphia (Fig. R1-7). This is because the Low-E window film has a low IR emissivity, thus effectively preventing heat dissipation. For example, in Philadelphia, Low-E film can save more energy than the MTPW and tinted film in winter from November to March due to the reduction of energy usage in the heating system. In the hot seasons from April to October, the Low-E film can still save energy due to the low NIR transmittance thus blocking part of solar radiation. But it does not perform remarkable energy-saving ability compared with the tinted film and MTPW film (Fig. R1-8C and D). This phenomenon is especially obvious in hot areas (e.g., Orlando and Honolulu), because Low-E window film suppresses the heat brought by solar radiation dissipating from the indoor side to the outdoor side, thus leading to high energy consumption for cooling systems in subtropical and tropical areas (Fig. R1-8E and F). Therefore, in these areas, our MTPW film is a better choice.

As mentioned before, due to the light modulation region of the thermochromic perovskite being

in the visible light range, it can also be conveniently integrated with Low-E coating (i.e. functional region in the IR range) to combine the advantages of these two coatings and develop more advanced multi-layer window films for higher energy saving abilities. In our previously published work, we indeed developed a window with multi-coating structures, including visible-light modulated thermochromic perovskite and IR-reflected Low-E layer, to maximize the energy-saving potential for thermochromic smart windows. [R1-1]. We appreciate the reviewer's insightful comments, and the energy simulation part has been revised in the manuscript, and it is also provided in *Italics* below for convenience. Thank you so much.

Fig. R1-6. Photographs of (A) a commercial energy-saving tinted window film and (B) a commercial energy-saving Low-E window film (C) Transmittance spectra of the commercial window films and MTPW film. (D) Emissivity spectra of the commercial window films and MTPW film.

Fig. R1-7. Energy savings over a year obtained by using the MTPW, the tinted window film and the Low-E window film compared with a normal window without window films.

Fig. R1-8. (A) Locations of Philadelphia, Washington, Atlanta, Orlando and Honolulu as well

as the building model used in the EnergyPlus simulation. (B) Window structures (normal window, tinted window film, Low-E window film pasted on the window, and MTPW window film pasted on the window) in the EnergyPlus simulation. (C) Monthly energy consumption and savings in Philadelphia using different windows. (D) Energy consumption by types in Philadelphia using different windows. (E) Monthly energy consumption and savings in Orlando using different windows. (F) Energy consumption by types in Orlando using different windows.

Reference:

[R1-1] Liu, S., Li, Y., Wang, Y., Yu, K. M., Huang, B., & Tso, C. Y. (2022). Near-Infrared-Activated Thermochromic Perovskite Smart Windows. *Advanced Science*, **9**(14), 2106090.

Highlighted in yellow at Superhydrophobicity, Stability and Application of the MTPW:

The MTPW film was also compared with Low-E window films (Supplementary Fig. S23B) that have a low emissivity ($\epsilon=0.109$) in the long wave infrared region (Supplementary Fig. S23D). Due to the high thermal radiation reflectance, using Low-E film can achieve better thermal insulation for the indoor environment thus leading to even higher energy-saving performance than the MTPW film in those areas that have long and cold winters (Fig. 5G, Supplementary Fig. S24C and D). While, in hot areas or the summer time, the MTPW film demonstrates better energy-saving potential than the Low-E film (Fig. 5G, Supplementary Fig. S24E and F), since the high thermal reflection of Low-E film blocks the heat dissipation. These results suggest that the MTPW window film is promising for reducing HVAC energy consumption throughout the year, and it should be noted that due to the light modulation region of the T-Perovskite being in the visible light range, it can also be conveniently integrated with Low-E coating (i.e. functional region in the IR range) to develop more advanced multi-layer window films to achieve higher energy saving abilities¹⁶.

Revision at Supplementary Text 2:

To compare the energy-saving performance, EnergyPlus modeling was conducted for a commercial building using a commercial tinted film, Low-E film and the MTPW film in five cities of the U.S. (Supplementary Fig. S24A and B). The basic building information is listed in Supplementary Table S1, and the optical information of the normal glass window, the window with the tinted film, the Low-E film and the MTPW film used in the simulation was calculated via the WINDOW algorithm developed by the Lawrence Berkeley National Laboratory

(Supplementary Table S2). The climate information of each city is listed in Supplementary Table S3.

Taking Philadelphia as an example, the heating energy consumption in buildings with tinted window films was much higher than that in buildings with normal windows, especially in winter from November to March (Supplementary Fig. S24C), which eventually offset the saved cooling energy in hot weather (Supplementary Fig. S24D). In contrast, enabled by the smart thermally responsive color switching ability, the MTPW maintained a high solar heat gain in cold weather but a low solar transmittance in hot weather. Therefore, the heating demand when using the MTPW film did not significantly change compared to that when using the normal window in winter, whereas the cooling demand dramatically decreased in the transition seasons and in summer from April to October (Supplementary Fig. S24C), demonstrating the advantage of the smart optical regulation function. For the Low-E film with low IR emissivity, it has even better energy-saving potential than the MTPW film in winter from November to March. In the hot seasons from April to October, the Low-E film can still save energy due to the low NIR transmittance thus blocking part of solar radiation. But it does not perform remarkable energy-saving ability compared with the tinted film and MTPW film (Supplementary Fig. S24C and D). For hot areas (e.g., Orlando), the windows with the MTPW and tinted window film both showed better energy-saving ability than the normal window (Supplementary Fig. S24E and F), and the MTPW window exhibited better performance due to the lower solar transmittance at the hot state. However, Low-E window film suppresses the heat brought by solar radiation dissipating from the indoor side to the outdoor side, thus leading to low energy savings for cooling systems in subtropical and tropical areas.

4. Typo on line 78 “hinters” should be “hinders”

Response: Thank you so much for pointing out this typo. The spelling of “hinders” has been corrected (highlighted in yellow) in the revised manuscript, and the whole paper has also been double-checked by an English Editor to make sure that there are no other typos or grammatical mistakes. Thank you.

Reviewer #2:

Dear authors,

I would like to commend you on an impressive manuscript. I found it to be clear and quite extensive in terms of the amount of work which has been performed to address the various aspects of T-perovskites for smart window applications. There are, however, a few points which I believe should be addressed prior to publication:

Response: We sincerely appreciate your positive and valuable comments. We have taken careful note of your suggestions and have thoroughly addressed them. Please find our detailed response to your comments below. Thank you.

1. The improvements which you demonstrate are important and relevant for a potential application of such coatings for smart windows. However, I would like you to critically address some of the key characteristics of the solution which you propose. For one, you mention the importance of the aesthetic aspect of windows.

(1) The present solution displays a reddish color in the high temperature state. Red is typically not an attractive color in the architectural glass industry. Can anything be done to modify the color?

Response: Thank you so much for your questions. Perovskites are composed of organic cations (e.g. CH_3NH_3^+ (MA^+), $\text{HC}(\text{NH}_2)_2^+$ (FA^+), and Cs^+), metal cation (e.g. Pb^{2+} , Sn^{2+} , and Ge^{2+}) and halide ion (e.g. Cl^- , Br^- , and I^-). By carefully selecting the appropriate compositions, the band gap of halide perovskites can be precisely tuned, which in turn affects the color. We indeed investigated the influence of different halide elements on the color of thermochromic perovskites in our previously published work [R2-1]. We found that changing the halide component from iodine (I) to bromine (Br) or chlorine (Cl) leads to a larger bandgap, resulting in the color change of thermochromic perovskite from dark brown to light yellow in the hot states (Fig. R2-1). This optically tunable ability provides building occupants with more choices regarding window color and transmittance requirements. Moreover, we believe that more colors can be developed in future work by other strategies, such as element doping and structural coloring. Thank you!

Fig. R2-1: Photographs of the fabricated $\text{MA}_4\text{PbI}_{6-x-y}\text{Br}_x\text{Cl}_y \cdot 2\text{H}_2\text{O}$ perovskite thin films with color variations at the hot state. Reproduced with permission. [R2-1] Copyright 2022, Elsevier.

Reference:

[R2-1] Du, Y., Liu, S., Zhou, Z., Lee, H. H., Ho, T. C., Feng, S. P., & Tso, C. Y. (2022). Study on the halide effect of $\text{MA}_4\text{PbX}_6 \cdot 2\text{H}_2\text{O}$ hybrid perovskites—From thermochromic properties to practical deployment for smart windows. *Materials Today Physics*, **23**, 100624.

(2) Although the level of haze has been significantly decreased, 30% is still non-negligible.

Response: Thank you so much for your comment. The main goal of this study is to utilize a convenient and scalable solution coating method to achieve low haze and extend durability simultaneously for thermochromic perovskite smart windows. However, we agree with the reviewer that the haze of perovskite windows could be further reduced to achieve a clearer view for building occupants. Further reduction in haze can be achieved by optimizing the morphology of perovskite films. For this purpose, we are now indeed exploring other potential methods. One of the developments undergoing in our group is using the anti-solvent crystallization method to produce smooth and dense thermochromic perovskite thin films. Anti-solvents are nonpolar solvents with relatively low polarity (e.g. chlorobenzene), they can rapidly extract solvents (i.e. DMF) from the thermochromic perovskite precursor, leading to fast supersaturation of the perovskite precursor and precipitation in the fabrication process of perovskite films. This method enables significant improvement for the quality of perovskite morphology, thus leading to lower haze. We are now exploring various anti-solvents and seeking the most suitable one for thermochromic perovskite fabrication. We have obtained

some satisfactory preliminary results: the haze of thermochromic perovskite fabricated through the anti-solvent method in our group can reach ~10% for visible light (as shown in Fig. R2-2). However, the anti-solvent crystallization method requires a complicated operation technique and operation accuracy (e.g. anti-solvent dipping time, dipping location on the perovskite film etc.) during the spin coating process, which limits the scalability. In addition, the toxicity of the anti-solvent also needs to be considered. This work is still under development, and we hope we can further enhance the operability of the anti-solvent method. We have added discussions in the revised manuscript, and for your convenience, we have also provided them below in italics.

Thank you!

Fig. R2-2. The wavelength-dependent haze spectrum of thermochromic perovskite after the treatment of anti-solvents during the fabrication process.

Highlighted in yellow at Discussion:

In addition, there is still potential to further reduce the optical haze of MTPW. For instance, exploring encapsulation materials with a refractive index that matches T-Perovskite could further decrease RMS tolerance and subsequently reduce the haze. Another potential method is to utilize the anti-solvent crystallization technique to produce smoother T-Perovskite films. However, it is important to consider the toxicity of the anti-solvent and scalability issues associated with this approach.

2. Other window considerations:

(1) Water vapor is key to induce the TC effect. Can you comment on the minimum level of relative humidity for this type of window to work? In fact, what would happen if it was indeed encapsulated in a double-glazed window where the water vapor would be minimal.

Response: Thank you so much for your comments. The transition properties of thermochromic perovskites are closely linked to the relative humidity, and we have extensively investigated the relationship in our previous work [R2-2]. In general, the transition temperature decreases with the decrease in humidity. This implies that when the humidity is at an ultra-low level, the transition temperature could be even lower than the room temperature. Based on our previous study, when the humidity is below ~12%, the thermochromic perovskite consistently remains in the colored state, rendering the thermochromic function inactive. Here, we conducted an experiment for the MTPW to demonstrate the color switch with lower humidity . An MTPW was put in a humidity-controlled chamber in which the ambient temperature was around 18 °C and the relative humidity (RH) was reduced from 55% to 10%. As shown in Fig. R2-3, it was observed that the MTPW changed from a transparent state with a RH of 55% to a colored state with a RH of ~10% at the same temperature. This indicates that, when the humidity is lower than ~10%, the window is always in the colored state (i.e. losing thermochromism), even if the ambient temperature is only ~18 °C.

There are currently two types of double-glazed windows available in the market. The first type is the vacuum insulated window, which consists of two glass panes with a vacuum layer in between. Based on the experiment above, it is not suitable to seal thermochromic perovskite in vacuum windows because the ultra-low humidity within the vacuum layer would result in the loss of thermochromism. The second type is the general double-pane window, which has an air gap layer in the middle. When using general double-glazed windows to seal thermochromic perovskite, it is necessary to control the humidity level in the gap to prevent the loss of the thermochromism in low humidity conditions and to avoid damage to the perovskite in high humidity conditions. It is worth noting that the strength of this study lies in the proposal of a multicoating design to enhance durability while preserving the thermochromic properties of perovskite in ambient environments. Consequently, there is no need to seal the perovskite in double-glazed windows, which would otherwise require complex assembly techniques to control the humidity within the air gap. Most importantly, due to the extended durability and waterproof ability, our coatings can be directly deposited onto flexible PET substrates as window films, thus making the thermochromic perovskite glazing technique scalable and reducing retrofitting costs for buildings. Thank you!

Fig. R2-3. The MTPW transfers from the transparent state to the colored state with the decrease in humidity.

Reference:

[R2-2] Liu, S., Du, Y. W., Tso, C. Y., Lee, H. H., Cheng, R., Feng, S. P., & Yu, K. M. (2021). Organic hybrid perovskite ($\text{MAPbI}_{3-x}\text{Cl}_x$) for thermochromic smart window with strong optical regulation ability, low transition temperature, and narrow hysteresis width. *Advanced Functional Materials*, **31(26)** 2010426.

(2) Yes, the presented T-Perovskite has a high $\Delta\tau_{sol}$, but most of the optical change is actually happening in the visible. Can you comment on this? Can the change be increased in the NIR?

Response: Thank you so much for your comments. At present, perovskites demonstrate solar modulation primarily within the visible light spectrum, resulting in a constrained maximum $\Delta\tau_{sol}$ of approximately 25%. We concur with the reviewer's insightful suggestion regarding the exploration of broadband modulated perovskites spanning from visible light to NIR range, which presents a promising avenue for future investigations. The spectral response of perovskites is mainly determined by their bandgaps. One potential direction is the investigation of novel thermochromic perovskites with narrower band gaps in their colored state, enabling

the extension of absorption beyond visible light to NIR wavelengths. For example, the mixture of divalent organic cations with FA⁺ and MA⁺/Cs⁺ and in conjunction with divalent metal cations such as Pb²⁺ and Sn²⁺ for perovskite has shown promise in achieving a narrower band gap (e.g., the FA_{0.75}Cs_{0.25}Sn_{0.5}Pb_{0.5}I₃ perovskite exhibits a low bandgap of 1.2 eV) [R2-3, R2-4]. Nonetheless, a thorough investigation is required to assess the impact of such perovskite modifications on the thermochromic effect.

Another solution to this problem is composite materials. For example, vanadium dioxide (VO₂), possessing light modulation capabilities within the NIR range, can be combined with thermochromic perovskites. However, given the disparate optical and transition properties of these materials, careful consideration must be given to the design of such windows, ensuring the synergistic effects of material combinations on luminous transmittance, solar modulation and transition temperatures. Thank you again for your suggestion and we will continue the relevant research in our future study to further boost the optical modulation of perovskite smart windows. The related discussion was added in the revised manuscript and we have also provided them below in italics for your convenience.

Highlighted in yellow at Discussion:

Moreover, the current T-Perovskites exhibit optical modulation primarily in the visible light region. To enhance the solar modulation ability, it would be worthwhile to investigate new broadband-modulated perovskite materials that span from the visible light to NIR range. Possible approaches include tuning the bandgap of T-Perovskite or integrating T-Perovskite with other materials like VO₂.

References:

[R2-3] Chang, Z., Lu, Z., Deng, W., Shi, Y., Sun, Y., Zhang, X., & Jie, J. (2023). Narrow-bandgap Sn–Pb mixed perovskite single crystals for high-performance near-infrared photodetectors. *Nanoscale*, **15**(10), 5053-5062.

[R2-4] Eperon, G. E., Leijtens, T., Bush, K. A., Prasanna, R., Green, T., Wang, J. T. W., ... & Snaith, H. J. (2016). Perovskite-perovskite tandem photovoltaics with optimized band gaps. *Science*, **354**(6314), 861-865.

(3) Can you comment on the mechanical durability of the coating considering the top nanoparticle-based film?

Response: Thank you so much for your comments. In response to the reviewer's comments, we conducted an abrasion test on the SiO₂ nanoparticles. The coated side was placed face down on a sheet of 1500 grit sandpaper. Using a weight of 100 g, we longitudinally and transversely abraded the surface in both directions for a distance of 10 cm, which constituted one cycle of abrasion [R2-5]. The result shown in Fig. R2-4 demonstrates that the contact angle can be kept at ~160° for 20 cycles and ~130° for 50 cycles. Considering that windows are typically not exposed to harsh abrasion environments, the mechanical durability of the SiO₂ nanoparticles is sufficient for practical applications. We have included these results in the revised manuscript, and for your convenience, we have also provided them below in italics. Thank you so much for the suggestion.

Fig. R2-4. Water contact angle of SiO₂ nanoparticles coating after sandpaper abrasion.

Highlighted in yellow at Superhydrophobicity, Stability and Application of the MTPW:
In addition, the abrasion test for SiO₂ nanoparticles coating shows reliable hydrophobic functions for practical applications (Supplementary Fig. S20).

Highlighted in yellow at Contact Angle Measurements and Durability Test:
For the abrasion test, the coated side of the SiO₂ nanoparticles was placed face down on a sheet of 1500 grit sandpaper. Under a weight of 100 g, the sample was longitudinally and transversely abraded for a distance of 10 cm respectively, which constituted one cycle of abrasion.

Reference:

[R2-5] Chen, B., Zhang, R., Fu, H., Xu, J., Jing, Y., Xu, G., ... & Hou, X. (2022). Efficient oil-water separation coating with robust superhydrophobicity and high transparency. *Scientific Reports*, 12(1), 2187.

3. In the introduction, you mention: light-absorbing phase to a transparent phase as the temperature changes, which results in a color switch. Please clarify that the transparent phase is in the low temperature state.

Response: Thank you so much for your kind reminder. We have added it to the revised manuscript and provided it in Italics below for convenience.

Highlighted in yellow at Introduction:

The inherently low formation energy of thermochromic perovskites (T-Perovskites) enables the rapid transformation from a light-absorbing phase (hot state) to a transparent phase (cold state) as the temperature changes, which results in a color switch.

4. Bottlenecks: Points III and IV are very similar: high haze vs optical transparency. Could Point IV be the leakage of lead?

Response: Thank you so much for your kind suggestion. We acknowledge that there might be an overlap in the fundamental essence of bottlenecks (iii) and (iv). In accordance with the reviewer's comments, we revised the bottlenecks (iv) to lead leakage problems, as highlighted in yellow in the revised manuscript and provided it in Italics below for convenience. Thank you.

Highlighted in yellow at Design and Fabrication of the MTPW:

.....(iii) ultrahigh optical haze and insufficient optical transparency caused by poor surface morphology; and (iv) toxic lead leakage problems when encountering water.

5. You indicate that the PHOS consists in a SiO_x/SiON_x film; do you expect the refractive index to be slightly higher than 1.5 in this case? 1.55 as indicated further on?

Response: Thank you so much for your comments. To ensure the accuracy of the refractive index measurement of PHPS, we used another ellipsometer (J.A. Woollam Alpha-SE, measurement range: 400-900 nm) to confirm the *n* and *k* values again. Fig. R2-5 shows that the result is almost identical to our previous measurements (measured by J.A. Woollam RC2). It is noteworthy that prior investigations have demonstrated that the refractive index of SiON_x

typically falls within the range of 1.45 to 1.9 [R2-6, R2-7], thereby rationalizing the higher n value observed in the $\text{SiO}_x/\text{SiON}_x$ composite, exceeding 1.5. In addition, the extent of reaction in the $\text{SiO}_x/\text{SiON}_x$ highly depends on the curing conditions. Therefore, to make the description more precise, we emphasize that the n and k results of PHPS in Fig. 1D are specifically applicable to the PHPS film produced in our current study (i.e. curing at 100 °C for 3 hours). The manuscript has been revised accordingly, and we provided it in Italics below for convenience. Thank you so much for the comment.

Fig. R2-5. Complex refractive index of PHPS measured by RC2 and Alpha-SE.

Highlighted in yellow at Design and Fabrication of the MTPW:

The transmittance of the PHPS-coated glass was found to be ~90%, which is almost as high as that of the bare glass (Supplementary Fig. S3), and the complex refractive index of the fabricated PHPS in this study is shown in Fig. 1D.

Highlighted in yellow in the caption of Fig. 1D:

Complex refractive index of T-Perovskite in the cold and hot states as well as of the fabricated PHPS in this study.

References:

[R2-6] Kim, D., Jeon, G. G., Kim, J. H., Kim, J., & Park, N. (2022). Design of a flexible thin-film encapsulant with sandwich structures of perhydropolysilazane layers. *ACS Applied Materials & Interfaces*, **14**(30), 34678-34685.

[R2-7] Yin, L., Lu, M., Wielunski, L., Song, W., Tan, J., Lu, Y., & Jiang, W. (2012). Fabrication and characterization of compact silicon oxynitride waveguides on silicon chips. *Journal of Optics*, **14(8)**, 085501.

6. Have you tested thicker PHPS thicknesses than 1.8 μm ? Was the transition time judged too slow?

Response: Thank you so much for your comments. Based on the reviewer's suggestions, we expanded our investigation beyond the 1.8 μm thickness to explore the influence of varying PHPS thicknesses on the transition time. As shown in Fig. R2-6, the transition time in both the heating and cooling processes increases as the PHPS layer thickness increases. Remarkably, when the PHPS layer thickness reaches 2.28 μm , the transition times escalate to over 2.5 minutes and 7 minutes for the heating and cooling processes, respectively. The thicker PHPS can provide better protection for the T-Perovskite, but this inevitably prolongs the transition process. However, a transition time within several minutes should be acceptable for a building smart window, considering the relatively slow temperature fluctuation process in a day [R2-8] [R2-9]. Fig. R2-6 has been included in the supplementary materials as Fig. S8. Thank you very much for the comment.

Fig. R2-6. The transition (color switch) time of T-Perovskite upon heating cooling with different thicknesses of the PHPS coating.

References:

[R2-8] Ke, Y., Zhou, C., Zhou, Y., Wang, S., Chan, S. H., & Long, Y. (2018). Emerging thermal-

responsive materials and integrated techniques targeting the energy-efficient smart window application. *Advanced Functional Materials*, **28(22)**, 1800113.

[R2-9] Wu, S., Sun, H., Duan, M., Mao, H., Wu, Y., Zhao, H., & Lin, B. (2023). Applications of thermochromic and electrochromic smart windows: Materials to buildings. *Cell Reports Physical Science*.

7. You have quantified the level of roughness of your T-Perovskite films at 208 nm. Fig.1 E shows a geometrical optics representation for the presence of haze which can be misleading; indeed, one is most probably in the Mie scattering regime for visible wavelengths.

Response: Thank you so much for your comments. Considering the scale of the roughness, we agree with the reviewer that Mie scattering is proper to describe the light interaction process on the rough surface. Therefore, we revised that figure to avoid misleading the readers. The revised figure is shown below and moved to the supplementary material as Fig. S1A. Thank you so much for the comment.

Fig. R2-7: Light transmission through a rough surface.

8. You indicate: “Notably, both the $\tau_{lum,cold}$ and $\tau_{lum,hot}$ of the MTPW were higher than those of the TPW because less light scattering occurred at the surface, and therefore, the amount of haze decreased.” This is most probably the result of a decrease in the reflectance, which can easily be measured on an integrating sphere. Note that the level of scattering can also impact the level of absorption.

Response: Thank you so much for your comments. We apologize for the ambiguous explanation here. We agree with the reviewer that the increased τ_{lum} of MTPW is attributed to the decreased reflectance compared with the TPW. And following the reviewer’s instruction, we measured the reflection spectra of these two windows by using the UV-VIS-NIR

spectrometer (with 150 mm integrating sphere). The results as shown in Fig. R2-7 prove that the MTPW has a lower reflectance than the TPW due to the reduction in diffused reflection caused by light scattering. The calculation results indicate that MTPW exhibits a transmittance of 83.4%, reflectance of 9.1%, and absorptance of 7.5%, while TPW shows a transmittance of 78.2%, reflectance of 14.8%, and absorptance of 7.0%. The absorptance of MTPW and TPW are almost the same. The sentence in the manuscript, “Notably, both the $\tau_{lum,cold}$ and $\tau_{lum,hot}$ of the MTPW were higher than those of the TPW because less light scattering occurred at the surface, and therefore, the amount of haze decreased”, aims to introduce the investigation of haze reduction. However, we also recognize that this logic might have been presented abruptly, potentially leading to misconceptions among readers. Therefore, we revised the sentence to mention the reflectance reduction first before the haze reduction. The revision is highlighted in yellow in the revised manuscript and we have also provided it in Italics below for convenience. Fig. R2-8 has been added in the revised supplementary materials as Fig. S11. Thank you very much for the comment.

Fig. R2-8. The reflectance of TPW and MTPW.

Highlighted in yellow in the section of Optical Performance and Phase Transition Properties of the MTPW:

Notably, both the $\tau_{lum,cold}$ and $\tau_{lum,hot}$ of the MTPW were higher than those of the TPW because of the decreased reflectance (Supplementary Fig. S11). The key reason for the lower reflectance is less light scattering at the surface causing a reduction in hazing.

9. Equation (2) indicates that when the two media have closer refractive indices, the tolerance to the RMS is larger. As such, have you considered to encapsulate the T-Perovskite layer

within a higher refractive index material (e.g.: SiN which would be index matching)

Response: Thank you so much for your comments. From the target of reducing haze, we agree with the reviewer that a higher refractive index material can further increase the RMS tolerance of the thermochromic perovskite. However, considering the high sensitivity of perovskite, the feasibility of depositing the encapsulation material in a practical fabrication process should also be carefully considered. As we discussed in the section of Design and Fabrication of the MTPW, the deposition process should not have any adverse effects on thermochromic perovskite film. For example, we tried to coat the SiO₂ on the thermochromic perovskite by using the physical vapor deposition method (magnetron sputtering). Many black dots were found on the perovskite film (Fig. R2-9), implying that high-energy plasma during the sputtering process damaged the thermochromic perovskite. Similarly, the deposition of silicon nitride also generally relies on these high-vacuum coating methods [R2-10, R2-11], which could also damage the perovskite film. Another key consideration is that the coating method should have great scalability and low cost. The coating of PHPS/SiO₂ particles relies on the solution-based coating method (e.g. spin coating, spray coating), therefore, the whole fabrication process of this smart window can be conveniently scalable, this is especially important for fabricating window films for future applications. The reviewer's insightful suggestion for enhancing the optical properties is greatly appreciated, we will further explore more suitable encapsulation materials that have a matching refractive index with perovskite in future work. The related discussion was added in the manuscript and we have also provided it in Italics below for convenience.

Fig. R2-9. (A) SEM image of sputtering SiO₂ on the T-Perovskite. The rough surface implies that the sputtering method is unable to improve the surface morphology. (B) Photos of the T-Perovskite after the magnetron sputtering. These fadeless brown spots on its surface indicates that the high-energy plasma during the sputtering process can damage the T-Perovskite.

References:

[R2-10] Schmidt, S., Hanninen, T., Goyenola, C., Wissting, J., Jensen, J., Hultman, L., ... & Hogberg, H. (2016). SiN x coatings deposited by reactive high power impulse magnetron sputtering: process parameters influencing the nitrogen content. *ACS Applied Materials & Interfaces*, **8(31)**, 20385-20395.

[R2-11] Signore, M. A., Sytchkova, A., Dimaio, D., Cappello, A., & Rizzo, A. (2012). Deposition of silicon nitride thin films by RF magnetron sputtering: a material and growth process study. *Optical Materials*, **34(4)**, 632-638.

Highlighted in yellow at Discussion:

In addition, there is still potential to further reduce the optical haze of MTPW. For instance, exploring encapsulation materials with a refractive index that matches T-Perovskite could further decrease RMS tolerance and subsequently reduce the haze. Another potential method is to utilize the anti-solvent crystallization technique to produce smoother T-Perovskite films. However, it is important to consider the toxicity of the anti-solvent and scalability issues associated with this approach.

10. "Moreover, the n value of PHPS is between those of air and the T-Perovskite, and it can also serve as an antireflection layer to improve the total transmittance." In order for the film to be antireflective, a specific thickness would be required. The drop in reflection after the addition of the PHSP is most probably due to the lower amount of haze more than to an antireflective effect which would entail interference phenomena. Did the transmission or reflection change with the different thicknesses of PHSP?

Response: Thank you so much for your comments. We apologize for the inaccurate expression here. As the reviewer's comment, the anti-reflection effect only happens at a specific thickness for the anti-reflection layer, and this effect is normally applied to smooth and uniform coatings. However, considering the large roughness of the thermochromic perovskite layer, the anti-reflection theory is not suitable to explain the transmittance improvement. We agree with the reviewer that haze reduction contributes to the improvement of transmittance. And based on the reviewer's suggestion, we measured the transmittance and haze of the window by changing the thickness of PHPS. As shown in Fig. R2-10, the transmittance increases and the haze decreases with the increase of the PHPS thickness. Related discussions have been revised in the manuscript and figures have been added in the supplementary materials as Fig. S9. We have

also provided it in Italics below for convenience.

Fig. R2-10. (A) The transmittance variation by coating different thicknesses of PHPS on the T-Perovskite. (B) The haze variation by coating different thicknesses of PHPS on the T-Perovskite. (C) SEM image of 1.8 μm PHPS-coated T-Perovskite. The flat surface indicates uniform coverage. (D) SEM image of 0.3 μm PHPS-coated T-Perovskite. The rough surface indicates poor coverage.

Highlighted in yellow at Design and Fabrication of the MTPW:

The thickness of the PHPS layer influences the protection ability, optical performance, and transition time of the smart window. Note that while a thinner layer of PHPS results in a faster color switch process (Supplementary Fig. S8), the poor surface coverage on the T-Perovskite leads to weaker protection ability and poor optical performance (i.e. low transmittance and high haze) (Supplementary Fig. S9).

11. The FDTD study is interesting but the results are expected (especially for Figure 3A). As a result, I feel that the section on the reduction in haze is lengthy and could be shortened.

The BSDF data is interesting, however.

Response: Thank you so much for your suggestions. Based on the reviewer's comments, we have deleted part of the content in the FDTD study. And according to the format guideline of Nature Communication, we have to further shorten the length of the main text in the manuscript and moved part of the content/figures to the supplementary material to make the whole paper more compactable and readable. The revision part has been highlighted in yellow in the revised manuscript. Thank you so much for the comment.

12. You mention a commercial energy-saving window film; this is quite vague. Why was this particular film chosen?

Response: Thank you so much for your comments. The thermochromic functional region of the MTPW film is mainly in the visible light region, so we chose a visibly-tinted window film (Fig. R2-11A) in the building energy simulation to demonstrate the strength of the color switch film for year round energy savings. Based on the reviewer's comment, to make the energy-saving simulation more convincing, we also added another widely used commercial window film: Low-E film in the simulation for comparison. It should be noted that different from the MTPW film and the tinted window film, Low-E window film focuses on reducing heat transfer through low emittance (i.e high IR reflection). Therefore, Low-E film is especially suitable for cold areas to mitigate heat loss (i.e. thermal radiation from indoor to outdoor) in winter.

As shown in Fig. R2-11B, we chose the commercial Low-E window film that has high luminous transmittance (84%, Fig. 2-11C) but a low emissivity of 0.109, which is much lower than tinted and MTPW window films whose emissivity are larger than 0.9 (Fig. 2-11D). The simulation in five cities of the U.S. (Fig. R2-13A) shows that Low-E window film has a better energy-saving potential than the MTPW in some areas that have long and cold winters, such as Washington and Philadelphia (Fig. R2-12). This is because the Low-E window film has a low IR emissivity, thus effectively preventing heat dissipation. For example, in Philadelphia, Low-E film can save more energy than the MTPW film in winter from November to March due to the reduction of energy usage in the heating system. In the hot seasons from April to October, the Low-E film can still save energy due to the low NIR transmittance thus blocking part of solar radiation. But it does not perform remarkable energy-saving ability compared with the tinted film and MTPW film (Fig. R2-13C and D). This phenomenon is especially obvious in hot areas (e.g., Orlando and Honolulu), because Low-E window film suppresses the heat brought by solar radiation dissipating from the indoor side to the outdoor side, thus leading to high energy consumption for cooling systems in subtropical and tropical areas (Fig R2-13E and F). Therefore, in these areas, our MTPW film is a better choice.

As mentioned before, due to the light modulation region of the T-Perovskite being in the visible light range, it can also be conveniently integrated with Low-E coating (i.e. functional region in the IR range) to combine the advantages of these two coatings and develop more advanced multi-layer window films for higher energy saving abilities. In our previously published work, we indeed had developed a window with multi-coating structures, including visible-light modulated T-Perovskite and IR-reflected Low-E layer, to maximize the energy-saving potential for thermochromic smart windows. [R1-12]. We appreciate the reviewer's insightful comments,

and the energy simulation part has been revised in the manuscript, and it is also provided in Italics below for convenience. Thank you so much.

Fig. R2-11. Photographs of (A) a commercial energy-saving tinted window film and (B) a commercial energy-saving Low-E window film (C) Transmittance spectra of the commercial window films and MTPW film. (D) Emissivity spectra of the commercial window films and MTPW film.

Fig. R2-12. Energy savings over a year obtained by using the MTPW, the tinted window film and the Low-E window film compared with a normal window without window films.

Fig. R2-13. (A) Locations of Philadelphia, Washington, Atlanta, Orlando and Honolulu as well as the building model used in the EnergyPlus simulation. (B) Window structures (normal window, tinted window film, Low-E window film pasted on the window, and MTPW window film pasted on the window) in the EnergyPlus simulation. (C) Monthly energy consumption and savings in Philadelphia using different windows. (D) Energy consumption by types in Philadelphia using different windows. (E) Monthly energy consumption and savings in Orlando using different windows. (F) Energy consumption by types in Orlando using different windows.

Reference:

[R1-12] Liu, S., Li, Y., Wang, Y., Yu, K. M., Huang, B., & Tso, C. Y. (2022). Near-Infrared-Activated Thermochromic Perovskite Smart Windows. *Advanced Science*, **9**(14), 2106090.

Highlighted in yellow at Superhydrophobicity, Stability and Application of the MTPW:

The MTPW film was also compared with Low-E window films (Supplementary Fig. S23B) that have a low emissivity ($\epsilon=0.109$) in the long wave infrared region (Supplementary Fig. S23D). Due to the high thermal radiation reflectance, using Low-E film can achieve better thermal insulation for the indoor environment thus leading to even higher energy-saving performance than the MTPW film in those areas that have long and cold winters (Fig.5G, Supplementary Fig S24C and D). While, in hot areas or the summer time, the MTPW film demonstrates better energy-saving potential than the Low-E film (Fig.5G, Supplementary Fig S24E and F), since the high thermal reflection of Low-E film blocks the heat dissipation. These results suggest that the MTPW window film is promising for reducing HVAC energy consumption throughout the year, and it should be noted that due to the light modulation region of the T-Perovskite being in the visible light range, it can also be conveniently integrated with a Low-E coating (i.e. functional region in the IR range) to develop more advanced multi-layer window films to achieve higher energy saving abilities¹⁶.

Revision at Supplementary Text 2:

To compare the energy-saving performance, EnergyPlus modeling was conducted for a commercial building using a commercial tinted film, Low-E film and the MTPW film in five cities of the U.S. (Supplementary Fig. S24A and B). The basic building information is listed in Supplementary Table S1, and the optical information of the normal glass window, the window with the tinted film, the Low-E film and the MTPW film used in the simulation was calculated via the WINDOW algorithm developed by the Lawrence Berkeley National Laboratory (Supplementary Table S2). The climate information of each city is listed in Supplementary Table S3.

Taking Philadelphia as an example, the heating energy consumption in buildings with tinted window films was much higher than that in buildings with normal windows, especially in winter from November to March (Supplementary Fig. S24C), which eventually offset the saved cooling energy in hot weather (Supplementary Fig. S24D). In contrast, enabled by the smart thermally responsive color switching ability, the MTPW maintained a high solar heat gain in cold

weather but a low solar transmittance in hot weather. Therefore, the heating demand when using the MTPW film did not significantly change compared to that when using the normal window in winter; whereas the cooling demand dramatically decreased in the transition seasons and in summer from April to October (Supplementary Fig. S24C), demonstrating the advantage of the smart optical regulation function. For the Low-E film with low IR emissivity, it has even better energy-saving potential than the MTPW film in winter from November to March. In the hot seasons from April to October, the Low-E film can still save energy due to the low NIR transmittance thus blocking part of solar radiation. But it does not perform remarkable energy-saving ability compared with the tinted film and MTPW film (Supplementary Fig. S24C and D). For hot areas (e.g., Orlando), the windows with the MTPW and tinted window film both showed better energy-saving ability than the normal window (Supplementary Fig. S24E and F), and the MTPW window exhibited better performance due to the lower solar transmittance at the hot state. However, Low-E window film suppresses the heat brought by solar radiation dissipating from the indoor side to the outdoor side, thus leading to low energy savings for cooling systems in subtropical and tropical areas.

13. Outstanding optical properties is a bit strong. In comparison to traditional T-perovskites perhaps, but in comparison to the overall demands for window coatings, there is still room for improvement.

Response: Thank you so much for your comments. We are aware that some words are a bit strong to describe the performance of the window. We thoroughly reviewed the manuscript and refined our choice of words to avoid absolutes and adopted a more measured tone. The revision is highlighted in yellow in the revised manuscript and we have also provided it in Italics below for convenience. Thank you so much for the comment.

Highlighted in yellow at Abstract:

The MTPW demonstrates ~~outstanding~~ superhydrophobicity and maintains a solar modulation ability above 20% during a 45-day aging test....

Highlighted in yellow at Introduction:

Therefore, developing durable and water-repellent T-Perovskite windows with ~~outstanding~~ good optical and transition properties is imperative and urgent.

Highlighted in yellow at Discussion:

In this study, inspired by the structure of a mask, a TPW with ~~outstanding~~ improved optical properties and enhanced durability was explored.

14. Please specify the observer (CIE 1931 or 1964) for the luminance transmittance calculations.

Response: Thank you so much for your comments. CIE 1931 was used as the observer for the luminance transmittance calculations [R2-13]. We have added it at the Characterization in the revised manuscript and provided it in Italics below for convenience. Thank you for the comment.

Highlighted in yellow at Characterization:

The luminous transmittance (τ_{lum}) of each window was calculated based on CIE (International Commission on Illumination) 1931 standard observer by $\tau_{lum} = \frac{\int_{\lambda=380nm}^{780nm} \bar{y}(\lambda)\tau(\lambda)d\lambda}{\int_{\lambda=380nm}^{780nm} \bar{y}(\lambda)d\lambda}$ to quantify the amount of transmitted visible light, where $\tau(\lambda)$ is the transmittance of the window at wavelength λ . $\bar{y}(\lambda)$ is the photopic luminous efficiency of the human eye.

Reference:

[R2-13] Oleari, C. (2015). Standard colorimetry: definitions, algorithms and software. John Wiley & Sons.

15. The manuscript is well written, but I have found a few instances where it could be improved:

- In the abstract: Inspire by the medical mask structure. I would say inspired by the structure of medical masks.
- Hinters should be hindere.
- At both the cold and hot states should be in both the cold and hot states.
- “medical masks have become important personal belongings”; please rephrase.
- to block most of the virus“es”
- we select“ed” a widely used inorganic oxide
- SiO₂ coating normally relies on; SiO₂ coating“s are typically deposited”...
- 100 °C for 1 hour in “a” glove box to form a 1.6 μm-thick T-Perovskite film
- Figure 1: Repels water droplet “s”.
- Rayleigh roughness criteri"um"

-
- Conclusion (not discussion)
 - Enthalpy testing chamber? Are you sure enthalpy is the right term here?

Response: Thank you so much to point out these typos and improper expressions. The inaccurate spellings have been corrected (highlighted in yellow) in the revised manuscript, and the whole paper has also been double-checked by an English Editor to make sure that there are no other typos or grammatical mistakes. And we have rephrased “medical masks have become important personal belongings” to “medical masks have been widely used to prevent viral transmission”. In addition, according to the formatting instructions of Nature Communications, the last part of the article is “Discussion”. Therefore, we followed the journal’s instructions and did not use the “Conclusion” title. Thank you for your understanding. For the name of “enthalpy testing chamber”, we are aware that this name may mislead the readers. So we have revised it to “environmental test chamber”, which is a more common name for temperature/humidity control instruments. Thank you so much for the comments.

REVIEWER COMMENTS

Reviewer #1 (Remarks to the Author):

The authors have nicely addressed all my concerns, and I congratulate them on an impressive scientific study. I support publication.

Reviewer #2 (Remarks to the Author):

Dear authors,

Thank you for taking the time to address the reviewers' comments in such a rigorous and diligent way.

I have only two final comments to address:

- Line 316 still mentions the antireflection effect.

- When considering the EnergyPlus simulations, you indicate in Fig. S24 that all films were placed on the interior surface of a double-pane window. I understand why this is the case for your coating, as you clearly explained. However, this is typically not the case for a low-e coating which is usually placed on either one of the interfaces within the gap which is filled with a lower thermal conductivity gas such as argon. It would be interesting to run your simulations once more in this closer to reality configuration.

Thank you and best regards

Research Article, No. NCOMMS-23-15053A

Title: Mask-Inspired Breathable and Durable Thermochromic Perovskite Smart Windows

Response Statements

Dear Editor and Reviewers,

I would like to express my sincere gratitude to the editor and reviewers for dedicating time to review this revised manuscript. We have carefully considered the additional two comments and provided responses in a point-by-point manner. The revised sections in the manuscript are highlighted in yellow. Thank you so much for your valuable comments and support.

Reviewer #1:

The authors have nicely addressed all my concerns, and I congratulate them on an impressive scientific study. I support publication.

Response: Thank you for your positive feedback and support. We deeply appreciate the time and effort you dedicated to reviewing our manuscript. Thank you very much again for your support of our work.

Reviewer #2:

Dear authors,

Thank you for taking the time to address the reviewers' comments in such a rigorous and diligent way. I have only two final comments to address:

Response: We highly appreciate your positive and valuable comments. Your feedback has been truly helpful in improving our work. Please find our detailed response to your comments below.

1. Line 316 still mentions the antireflection effect.

Response: Thank you so much for your comment. We sincerely apologize for the improper expression in Line 316. In accordance with the reviewer's feedback, we have revised the sentence by removing the mention of the antireflection effect. The revised sentence is provided below. Thank you.

Highlighted in yellow at Optical Performance and Phase Transition Properties of the MTPW:

In short, we conclude that the top PHPS layer significantly enhances the optical performance through a large reduction in the haze and an increase in the total transparency by smoothing the originally rough surface.

2. When considering the EnergyPlus simulations, you indicate in Fig. S24 that all films were placed on the interior surface of a double-pane window. I understand why this is the case for your coating, as you clearly explained. However, this is typically not the case for a low-e coating which is usually placed on either one of the interfaces within the gap which is filled with a lower thermal conductivity gas such as argon. It would be interesting to run your simulations once more in this closer to reality configuration.

Response: Thank you so much for your comment. Considering the convenience of installation and cost during the actual process of retrofitting a building, flexible commercial window films are typically designed by the manufacturer to be directly pasted to the interior surface of the existing window (including commercial Low-E window films). This is the main reason why the Low-E film was placed on the interior surface of the window in our simulation. Additionally, in order to ensure a fair comparison among the different window films in the EnergyPlus simulation, we positioned all window films in the same position. We understand the reviewer's comment regarding the monolithic Low-E window that is directly assembled in the factory, where the Low-E coating is applied to the interfaces within the argon-filled gap to protect the coating. To address this concern, we conducted an additional simulation to demonstrate this scenario. As illustrated in Fig. R2-1A below, we compared the situation when the Low-E coating is applied to different positions on the window. The results in Fig. R2-1B indicate that the energy consumption using type 1 and type 2 window configurations only exhibit slight differences. This is due to the small temperature difference between the glass and air, resulting in an insignificant variance in thermal radiation reflectance caused by the Low-E coating for both types 1 and 2. It should be noted that the primary objective of the final part of the manuscript is to emphasize that the newly proposed thermochromic perovskite coating in this study can be designed as a flexible window film, enabling convenient application on existing windows. Therefore, we employed the type 1 window configuration in the EnergyPlus simulation to facilitate a comparison with perovskite window film. We sincerely appreciate the suggestions provided by the reviewer and hope that the above response adequately addresses the concerns raised. Thank you for your valuable comment.

Fig. R2-1. (A) Schematic illustrating the placement of the Low-E coating on different sides of the window. (B) Energy consumption comparison between the two types of Low-E coated windows.

REVIEWERS' COMMENTS

Reviewer #2 (Remarks to the Author):

The authors have resolved the two minor issues I had highlighted. The manuscript can be published.

Thank you and best regards

Research Article, No. NCOMMS-23-15053B

Title: Mask-Inspired Moisture-transmitting and Durable Thermochromic Perovskite Smart Windows

Response Statements

Reviewer #2:

The authors have resolved the two minor issues I had highlighted. The manuscript can be published.

Response: Thank you very much for your positive feedback and support. We sincerely appreciate the time and effort you dedicated to reviewing our manuscript.